



**Time-Of-Flight monitoring reveals higher sediment redistribution rates related to burrowing animals**
**than previously assumed**
*Paulina Grigusova[1], Annegret Larsen[2], Sebastian Achilles[1], Roland Brandl[3], Camilo del Río[4,5], Nina Farwig[6],*
*Diana Kraus[6], Leandro Paulino[7], Patricio Pliscoff[4,8,9], Kirstin Übernickel[10], Jörg Bendix[1]*
[1] Laboratory for Climatology and Remote Sensing, Department of Geography, University of Marburg, 35037
Marburg, Germany; paulina.grigusova@staff.uni-marburg.de (P.G.); bendix@geo.uni-marburg.de (J.B.)
[2] Soil Geography and Landscape, Department of Environmental Sciences,
Wageningen University & Research, 6700 AA Wageningen, The Netherlands; annegret.larsen@wur.nl
[3] Animal Ecology, Department of Biology, University of Marburg, 35032 Marburg, Germany;
brandlr@biologie.uni-marburg.de
[4] Facultad de Historia, Geografía y Ciencia Política, Instituto de Geografía, Pontificia Universidad Católica de
Chile, 782-0436 Santiago, Chile; pliscoff@uc.cl; cdelriol@uc.cl
[5] Centro UC Desierto de Atacama, Pontificia Universidad Católica de Chile, 782-0436 Santiago, Chile;
cdelriol@uc.cl
[6] Conservation Ecology, Department of Biology, University of Marburg, 35047 Marburg, Germany;
diana.kraus@biologie.uni-marburg.de (D.K.); nina.farwig@biologie.uni-marburg.de (N.F.)
[7] Facultad de Agronomía, Universidad de Concepción, 3780000 Chillán, Chile; lpaulino@udec.cl
[8] Facultad de Ciencias Biológicas, Departamento de Ecología, Pontificia Universidad Católica de Chile,
8331150 Santiago, Chile; pliscoff@uc.cl
[9] Center of Applied Ecology and Sustainability (CAPES), Pontificia Universidad Católica de Chile, 8331150
Santiago, Chile; pliscoff@uc.cl
[10] Earth System Dynamics, Department of Geosciences, University of Tübingen, 72076 Tübingen, Germany;
kirstin.uebernickel@uni-tuebingen.de
*Corresponding author:*
Paulina Grigusova
paulina.grigusova@staff.uni-marburg.de





**Abstract**
Burrowing animals influence surface microtopography and hillslope sediment redistribution, but changes often
remain undetected due to a lack of autonomous high resolution field monitoring techniques. In this study we
present a new approach to quantify microtopographic variations and surface changes caused by burrowing
animals and rainfall-driven erosional processes applied to remote field plots in arid and mediterranean Chile.
We compared the mass balance of redistributed sediment within plot areas affected and not affected by
burrowing animals, quantified the cumulative sediment redistribution caused by animals and rainfall, and
upscaled the results to the hillslope scale. The new instrument showed a very good detection accuracy. The
cumulative sediment redistribution within areas affected by burrowing animals was higher (-10.44 cm$^3$ cm$^{-2}$ year$^{-1}$
) in the mediterranean than the arid climate zone ( -1.41 cm$^3$ cm$^{-2}$ year$^{-1}$). Daily sediment redistribution during
rainfall within areas affected by burrowing animals were up to 350% / 40% higher in the mediterranean / arid
zone compared to the unaffected areas, and much higher than previously reported in studies not based on
continuous microtopographic monitoring. Furthermore, 38% of the sediment eroding from the burrows
accumulated within the burrow entrance while 62% was incorporated into overall hillslope sediment flux. The
cumulative sediment excavation by the animals was 14.62 cm$^3$ cm$^{-2}$ year$^{-1}$ in the mediterranean and 16.41 cm$^3$
cm$^{-2}$ year$^{-1}$ in the arid climate zone. Our findings can be implemented into long-term soil erosion models that rely
on soil processes but do not yet include animal-induced surface processes on microtopographical scales in their
algorithms.

**Keywords:** Biogeomorphology, bioturbation, sediment transport, burrowing animals, rainfall, Time-of-Flight
camera, Chile







**Graphical abstract**

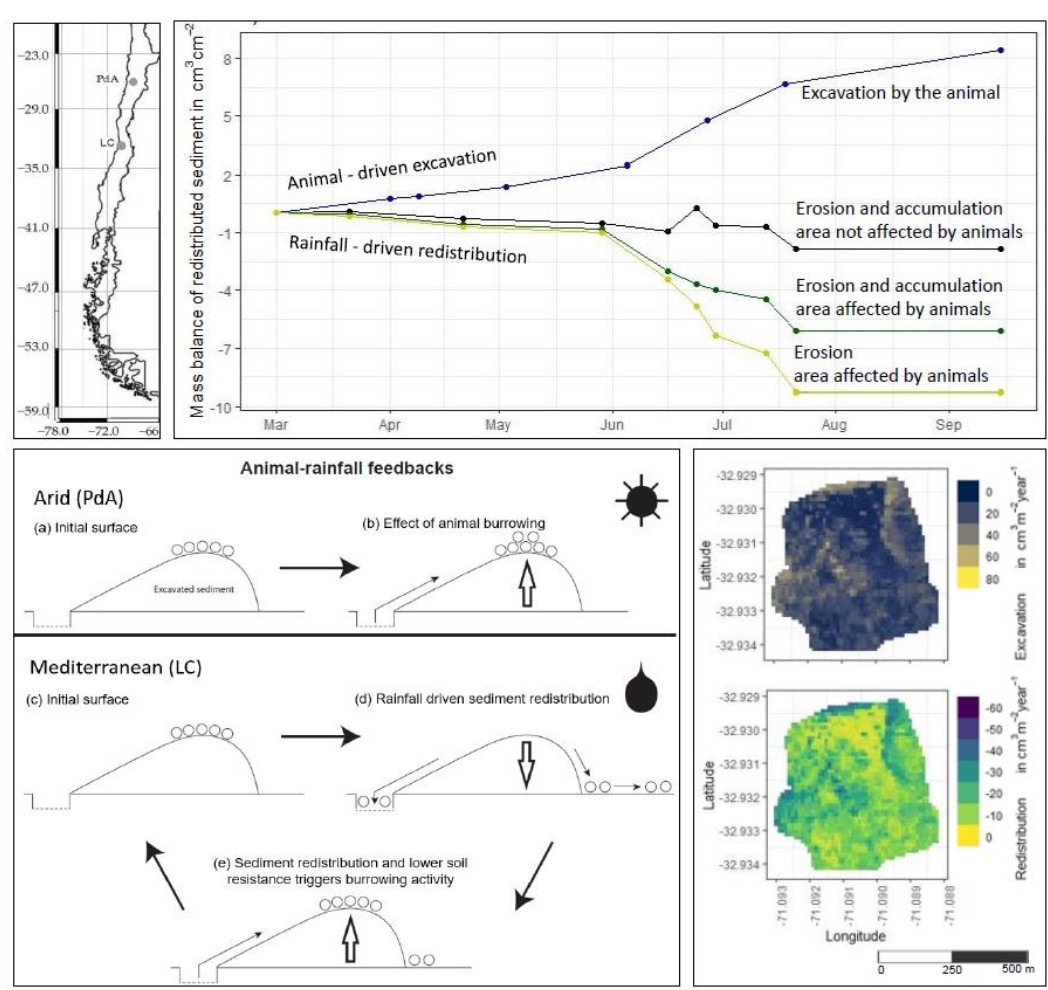





## 1. Introduction

Animal burrowing activity affects surface microtopography (Reichman and Seabloom, 2002; Kinlaw and Grasmueck, 2012), surface roughness (Yair, 1995; Jones et al., 2010; Hancock and Lowry, 2021) and soil physical properties (Ridd, 1996; Yair, 1995; Hall et al., 1999; Reichman and Seabloom, 2002; Hancock and Lowry, 2021; Coombes, 2016; Larsen et al., 2021). Previous studies estimated both positive as well as negative impacts of burrowing animals on sediment redistribution rates. The results were obtained by applying tests under laboratory conditions using rainfall simulators, conducting several field campaigns weeks to months apart, or by measuring the volume of excavated or eroded sediment in the field using methods such as erosion pins, splash boards, or simple rulers (Imeson and Kwaad, 1976; Reichman and Seabloom, 2002; Wei et al., 2007; Le Hir et al., 2007; Li et al., 2018; Li et al., 2019c; Li et al., 2019b; Voiculescu et al., 2019; Chen et al., 2021; Übernickel et al., 2021b; Li et al., 2019a). Although burrowing animals are generally seen as ecosystem engineers (Gabet et al., 2003; Wilkinson et al., 2009), their role in soil erosion, in general, and for numerical soil erosion models, in particular, is, to date, limited to predictions of the burrow locations and particle mixing at these locations (Black and Montgomery, 1991; Meysman et al., 2003; Yoo et al., 2005; Schiffers et al., 2011). The complex interaction of sediment excavation and accumulation and erosion processes at the burrow and hillslope scales are not yet included in the modelling, as for this, a suitable method capable of measuring all occurred redistribution processes is needed.

The reason for this knowledge gap is that previous studies have not provided data on low magnitude but frequently occurring sediment redistribution due to the specific limitations of their approaches. Field experiments with, for example, rainfall simulators can unveil processes but cannot cover the time-dependant natural dynamics of sediment redistribution. For data samplings that used methods such as erosion pins or splash boards, the sites had to be revisited each time and the data were thus obtained only sporadically (Imeson and Kwaad, 1976; Hazelhoff et al., 1981; Richards and Humphreys, 2010). Similarly, estimations of the excavated sediment volume are currently limited to one-time measurements or studies conducted several months apart (Black and Montgomery, 1991; Hall et al., 1999; Yoo et al., 2005). We expect that non-continuously conducted measurements do not include all frequently occurring excavation and erosion processes. For this, a spatio-temporally high-resolution and continuous monitoring of sediment redistribution is needed.

High-resolution, ground-based imaging sensing techniques might overcome such aforementioned problems. Terrestrial laser scanner systems have shown to be a suitable tool for estimation of sediment redistribution and erosion processes (Nasermoaddeli and Pasche, 2008; Afana et al., 2010; Eltner et al., 2016b; Eltner et al., 2016a; Longoni et al., 2016). However, they are expensive and labour-intensive and, thus, the study sites had to be personally revisited for each measurement. A continuous, autonomous monitoring of many mound areas in parallel is thus not possible. However, Time-of-Flight (ToF) technology offers new possibilities for a high-resolution monitoring of sediment redistribution (Eitel et al., 2011; Hänsel et al., 2016) but a cost-effective field monitoring device is still missing.

In our study, we developed, tested and applied a cost-effective Time-Of-flight camera to autonomously monitor the rainfall-driven and animal-driven sediment redistribution in areas affected by burrowing animals with high temporal (four times a day) and spatial (6 mm) resolution. For this, we equipped several plots in remote study sites of Chile in the arid and mediterranean climate zone. We selected these sites in order to analyse sediment redistribution by burrowing activity of vertebrates under different rainfall regimes and as these sites were particularly shown to be strongly affected by burrowing activity (Grigusova et al., 2021). Then,





we quantified the daily sediment redistribution within areas affected and not affected by burrowing animals.
We analysed the impacts of animal burrowing activity and rainfall on the sediment redistribution and quantified
the volume of sediment which is additionally incorporated to the hillslope sediment flux due to presence of
burrows. Finally, we estimated sediment redistribution on a burrow scale and upscaled sediment redistribution
rates to the entire hillslopes.

**2. Study sites**
Our study sites were located in the Chilean Coastal Cordillera in two climate zones (Fig. 1): in the National
Park Pan de Azúcar (further as Pan de Azúcar or PdA) and the National Park La Campana (further as La
Campana or LC). The Las Lomitas site in PdA is located in the arid climate zone of the Atacama Desert with a
precipitation rate of 12 mm year$^{-1}$, and it has a mean annual temperature of 16.8 °C (Übernickel et al., 2021a).
Here, the vegetation cover is below 5%, and it is dominated by small desert shrubs, several species of cacti
(*Eulychnia breviflora, Copiapoa atacamensis)* and biocrusts (Lehnert et al., 2018). LC is located in the
mediterranean climate zone with a precipitation rate of 367 mm year$^{-1}$ and a mean annual temperature of
14.1 °C (Übernickel et al., 2021a). LC is dominated by an evergreen sclerophyllous forest with endemic palm
trees, *Jubaea chilensis*. Both research sites have a granitic rock base, and the dominating soil texture is sandy
loam (Bernhard et al., 2018). In PdA, the study setup consisted of one north-facing and one south-facing
hillslope. The hillslope inclinations were ~20°, and a climate station was located ~15 km from the camera sites.
In LC, the setup consisted of two north-facing and one south-facing hillslopes. The hillslope inclinations were
~25°, and a climate station was located ~250 m from the south-facing hillslope (Übernickel et al., 2021a).

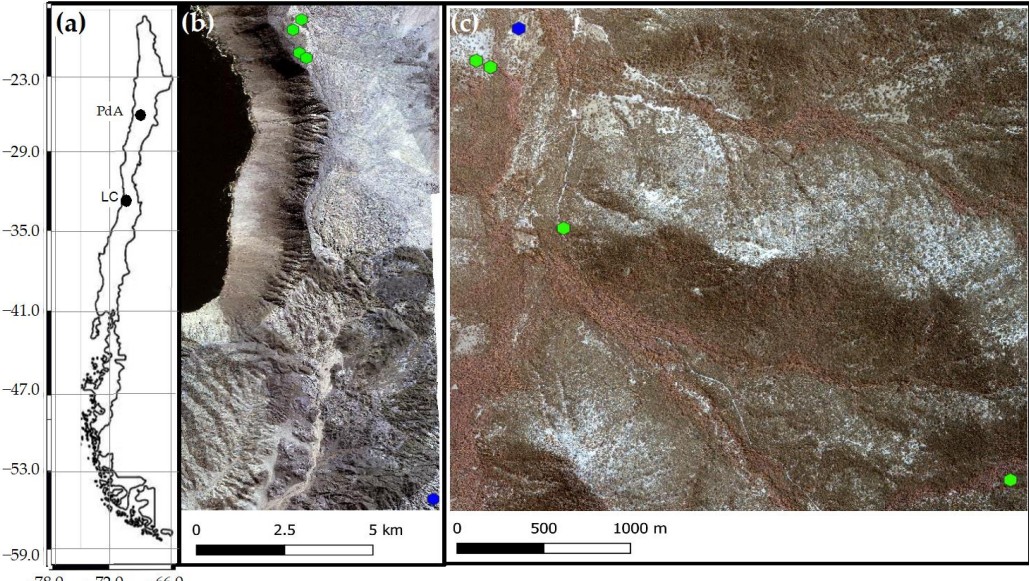

**Figure 1.** Location of the cameras and climate stations on which this study was based. Black points show the
location of the research sites in Chile. The green points represent the camera plots, and the blue points the
climate stations: (**a**) Location of study sites in Chile: PdA stands for Pan de Azúcar, LC for La Campana; (**b**)





Study setup in Pan de Azúcar; (**c**) Study setup in LC. The background images in (**b**) and (**c**) are orthophotos
created from WorldView-2 data from 19 July 2019. For exact latitude and longitude see Table A2.

**3. Methodology**

**3.1 Time-of-Flight (ToF) principle**

A Time-of-Flight-based camera illuminates an object with a light source, usually in a non-visible
spectrum, such as near-infrared, for a precise length of time. The here employed cameras were pulse-based,
meaning the light pulse was first emitted by the camera, then reflected from the surface, and finally measured
by the camera using two temporary windows. The first temporary window measures the incoming reflected
light while the light pulse is also still emitting from the camera. The second temporary window measures the
incoming reflected light when no pulse is emitting from the camera. The distance from the camera to the object
can then be calculated as follows:
$$d = \frac{1}{2} * c * t * \left(\frac{g_1}{g_1 + g_2}\right) \quad . \tag{1}$$
In Eq. (1), d (m) is the distance from the camera to the object, c (m s$^{-1}$) is the speed of light (299,792,458 m s$^{-1}$
), t (s) is the overall time of the illumination and measurement, $g_1$ is the ratio of the reflected photons to all
photons accumulated in the first window, and $g_2$ the ratio of the reflected photons to all photons accumulated
in the second window (Sarbolandi et al., 2018; Li, 2014).
The sensor in our camera came from Texas Instruments and the data scan contained information on
320 x 240 points. The camera field of view (FOV) and the spatial resolution of the scans depended on the
height of the camera above the surface. The distance was calculated for every point, and the object was saved
in binary format as a collection of 3D points with *x*-, *y*- and *z*-coordinates. The scans taken by the camera were
transformed from the binary format to a point cloud. Each point in the point cloud was assigned to an *x*-, *y*- and
*z*-coordinate. The coordinates were distributed within a three-dimensional Euclidian space, with the point at
the camera nadir being the point of origin. *x*- and *y*-coordinates describe the distance to the point of origin (m).
*z*-coordinate describes the distance (m) from the object to the camera. The lowest point of the scanned surface
thus has the highest *z*-coordinate value.

**3.2 Data processing**

The distortion caused by the hillslope and the camera angle was corrected for each point cloud as
follows:
$$z_{cor} = z_{uncor} - \tan(\alpha + \beta) * distance(y_1 - y_i) \quad . \tag{2}$$
In Eq. (2), $z_{cor}$ is the corrected distance (m) between the camera and surface (m), $z_{uncor}$ is the uncorrected *z*-
coordinate (m), α is the tilt angle of the camera (°), β is the surface inclination (°), and $y_i$ is the distance of the
point to the point of origin at the camera nadir (m). The most frequent errors were identified and treated as
follows. Due to the ambient light reaching the camera sensor, the *z*-coordinate values of some of the points
were incorrect (scattering error). To remove this error, a threshold value was calculated for each point cloud:
$$\Omega = mean_{zcor-coordinates} \pm sd_{zcor-coordinates} \quad . \tag{3}$$
In Eq. (3), Ω is the threshold value, $mean_{zcor-coordinate}$ is the average value, and $sd_{zcor-coordinate}$ is the standard
deviation of the corrected *z*-coordinates (m). Then all points with a *z*-coordinate above and below this value
were deleted. Point clouds with more than 50% of points above the threshold value Ω were also not considered
for further processing. A drift error occurred when the *z*-coordinate values of around one-third of the point



clouds decreased by several centimetres from one point cloud to another. Here, the average *z*-coordinate of
ten point clouds before and after the drift were calculated, and the difference was added to *z*-coordinates of
the points affected by the drift. The corrected height values were then transformed into a digital surface model
(DSM).

**200    3.3 Accuracy of the ToF cameras**

The accuracy of the ToF camera was tested under laboratory conditions by recreating similar surface

conditions as in the field (sloping surface, covered by sediment). An artificial mound using sediment extracted
from a riverbank in central Germany was used, mimicking a mound created by a burrowing animal. During the
test, the camera was installed 100 cm above the surface. The camera FOV was 3 m$^2$ and the scan spatial
resolution was 6 mm. The surface was scanned twice by the ToF camera. Then $100 - 450$ cm$^3$ of sediment
was manually extracted from the mound. The volume of the extracted sediment was measured by a measuring
cup. After extraction, the surface was again scanned twice by the camera. The experiment was repeated 45
times with varying amounts of extracted sediment. The scans were transformed to point clouds in VoxelViewer-
0.9.10, and the point clouds were corrected according to Eq. (2) and (3). The *z*-coordinates of the two point
clouds before and two point clouds after the extraction were averaged. The point clouds were then transformed
into DSMs, and the differences between the time steps were calculated. A scan was taken of a smooth surface,
and the standard deviation of its *z*-coordinates was calculated and saved as a threshold value. Solely, the
differences between the DSMs below this threshold (0.2 cm) were considered in the calculation of the detected
sediment extraction. The detected extracted sediment volume was then calculated for each experiment as
follows:
$Vol_{detected} = \sum_{p}^{1} (DSM_{before} - DSM_{after}) * res^2$         ,            (4)
In Eq. (4), $Vol_{detected}$ is the volume of the extracted sediment as detected by the camera (cm$^3$), p is the number
of pixels, $DSM_{before}$ (cm) is the DSM calculated from the scan taken before the extraction, $DSM_{after}$ (cm) is the
DSM calculated from the scan taken after the extraction, res (cm) is the resolution of the scan, which was 0.6
cm. To evaluate the camera's accuracy, the measured volume of the extracted sediment was compared to the
volume detected by the camera. The camera's accuracy was estimated between the detected volume and
measured volume as follows:
$MAE = \sum_{1}^{n} \frac{(Vol_{detected} - Vol_{measured})}{area}$         .            (5)
In Eq. (5), MAE (cm$^3$/cm$^2$) is the mean absolute error, n is the number of scans, $Vol_{measured}$ (cm$^3$) is the volume
of the extracted sediment measured by the measuring cup, and the area is the total surface area monitored
by the camera (cm$^2$).

**228    3.4 Installation of the cameras in the field**

We installed 8 custom-tailored ToF-based cameras on 4 hillslopes in two climate zones in areas

including visible signs of bioturbation activity (burrows) and areas without visible signs of bioturbation (Fig. 2).
The cameras were installed in LC on the north-facing upper hillslope (LC-NU), north-facing lower hillslope (LC-
NL), south-facing upper hillslope (LC-SU) and the south-facing lower hillslope (LC-SL); in PdA on the north-
facing upper hillslope (PdA-NU), north-facing lower hillslope (PdA-NL), south-facing upper hillslope (PdA-SU)
and south-facing lower hillslope (PdA-SL). The custom-tailored cameras were installed during a field campaign
in March 2019, the monitoring took place for seven months, and the data were collected in October 2019. The

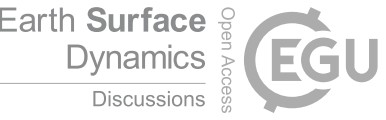

construction consisted of a 3D ToF-based sensor from Texas Instruments (Li, 2014), a RasperryPi single board
computer (SBC), a timer, a 12 V 12 Ah battery and three 20 W solar panels for unattended operation (Fig. 1).
Solar panels were located at the camera pole and were recharging the battery via a charge controller. The
camera was located approximately one meter above the surface, facing the surface with a tilt angle of 10
degrees. The timer was set to close the electric circuit 4 times a day: at 1 a.m., 5 a.m., 8 a.m. and 10 p.m. At
these times, the camera and the computer were turned on for 15 minutes. The camera turned on and took five
scans delayed one second from each other and sent them to the SBC. Each camera had its own WiFi (Wireless
Fidelity) and the data could be read from the SBC via Secure Shell (SSH).


**Figure 2.** Scheme and photo example of a Time-of-Flight-based camera installation in the field. The photo
example is from upper north-facing hillslope in La Campana. Black boxes describe single installation parts.
Purple descriptions are the variables needed for the correction of the scans. Roof, entrance and mound
describe areas affected by the burrowing animal. The $x$-, $y$- and $z$-coordinates are 3D coordinates identifying
the position of each point in space, where the $x$-coordinate is the length, $y$-coordinate is the width and the $z$-
coordinate is the distance between the camera sensor and the surface. α is the inclination of the camera, and
β is the surface inclination.

**3.5 Delineation of the Area Affected and Non-Affected by Burrowing Animals in the Camera's Field of**
**View**

The surface area scanned by the cameras was divided by a delineation scheme into areas affected
(A) and not (directly) affected (N) by burrowing animals. The affected areas included three sub-areas: (i) mound
(M), (ii) entrance (E) and (iii) burrow roof (R). "Mound" describes the sediment excavated by the animal while
digging the burrow. "Entrance" describes the entry to the animal burrow up to the depth possible to obtain via
the camera. "Burrow roof" describes the part of the sediment above and uphill the burrow entrance (Bancroft
et al., 2004). During the burrow's creation, sediment was not only excavated but also pushed aside and uphill
the entrance, which created the burrow's roof. We assume that this elevated microtopographical feature then



forms an obstacle for sediment transported from uphill, which leads to its accumulation in this area. The
remaining surface within the camera's FOV was classified as not affected (N) by the burrowing animal during
the creation of its burrow.
For the delineation, we used the DSM calculated from the point cloud, and a slope layer calculated
from the DSM (Horn, 1981). Entrance was assigned to an area determined by a search algorithm starting at
the lowest point of the DSM (pixel with the highest $z$-coordinate value). We increased the circular buffer around
the starting point by one pixel until the average depth of the new buffer points was not higher than the height
of the camera above the surface, or until the slope of at least 50% of the new buffer points was not 0. Then,
we masked all pixels within the buffer with a depth lower than the average depth of the points within the buffer,
which had a slope that was 0. The remaining pixels belonged to the entrance area. Then, the surface scan
was divided into an uphill and downhill part with regards to the entrance position. Both the uphill and the
downhill parts were subdivided into 16 squares.
To delineate the mound in the downhill part, we first identified the highest points (pixel with the lowest
$z$-coordinate value) within all 16 squares. We then calculated the distance of these maxima to the entrance,
and the pixel located nearest to the entrance was identified as the highest point of the mound (i.e., seed point).
Consecutively, we increased the circular buffer around the seed point by one pixel until the average depth of
the new buffer points was not lower than the height of the camera above the surface, or until the slope of at
least 50% of the new buffer points was not 0. Then, we masked all pixels within the buffer with a depth higher
than the average depth of the points within the buffer, which had a slope that was 0. The remaining pixels were
classified as mound area. To delineate burrow roof, we used the same approach as for the delineation of
mound and applied it on the uphill part of the surface scan. All pixels that were not classified during the entire
delineation process were treated as areas not affected by animals. The position and the boundaries of
entrance, mound and burrow roof were validated visually (Fig. 3 and A1).

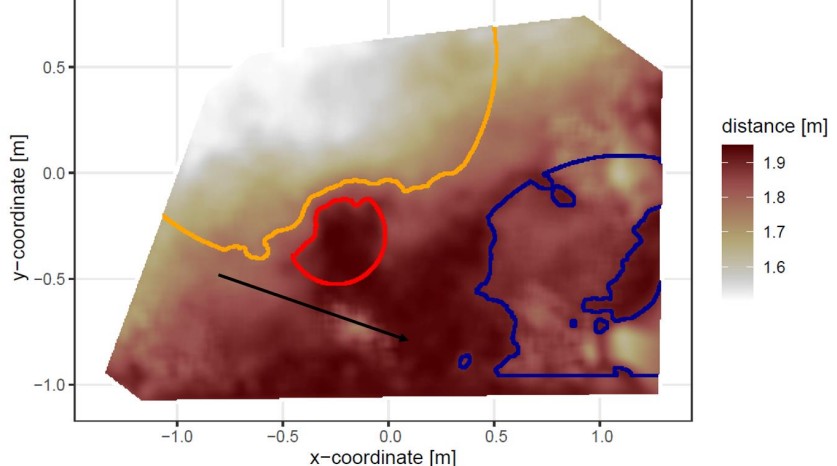

**Figure 3.** Corrected digital surface model of the camera on the upper north-facing hillslope in La Campana
with delineated areas. The point of origin of the coordinate system is at the camera nadir. Distance refers to
the distance between surface and camera. The red line delineates the burrow entrance, blue the mound and


Earth **Surface**
**Dynamics**
Discussions
EGU

orange the burrow roof. The area which was outside of any delineated area was classified as not affected by
animal burrowing activity. The arrow indicates a downhill direction of the hillslope.

**3.6 Calculation of daily sediment mass balance budget**

The volume of the redistributed sediment was calculated daily and was then cumulated from the first
day of monitoring. For the calculation of the daily sediment redistribution, the change in the surface level
detected by the camera was calculated first. For each day, the scans from the day before and after the
respective day were averaged and subtracted. As described in Section 2.2., all values with a difference below
and above the threshold value of 0.2 cm were set to 0. The redistributed sediment volume was then calculated
from the surface change for each pixel as follows:
$$Vol_{redistributed} = (S_b - S_a) * res^2 \qquad . \qquad\qquad (6)$$
In Eq. (6), $Vol_{redistributed}$ (cm³ pixel⁻¹) is the volume of the calculated redistributed sediment, $S_b$ (cm) the scan
before, $S_a$ (cm) the scan after the rainfall event and *res* is the spatial resolution (cm). Using the daily volume
of the redistributed sediment per pixel, we calculated the daily mass balance budget by summing the volume
of sediment eroding or accumulating within each delineated area.

**3.7 Calculation of animal-caused and rainfall-caused sediment redistribution**

The animal-caused sediment redistribution occurred when the animal actively reworked sediment
within its burrow. Under the assumption that the burrows are actively used by the animals, we defined four
cases when the sediment was redistributed due to the burrowing activity. For this, we pairwise compared the
DSMs of each scan with the scan saved before. The four cases were: (i) as the animal excavates sediment
from the entrance, the depth of the entrance must increase in the second scan; (ii) as the excavated sediment
accumulates on the mound, the height of the mound must increase in the second scan; (iii) as the burrowing
might lead to an expansion or a collapse of the burrow roof, an increase or decrease of the burrow roof must
occur between the scans; (iv) as the animal only digs within his burrow, no changes must occur between the
two scans within the area not affected by the animal. The animal-caused redistribution was then calculated for
these days as the volume of sediment redistributed within mound and burrow roof. The entrance was ignored
in the calculation. As the sediment excavated from the entrance accumulated on the mound and the sediment
accumulated within entrance collapsed from the burrow roof, by including the entrance in the calculation, these
sediment volumes would be counted twice.
The rainfall-caused sediment redistribution was calculated as follows: From the data from the climate
stations (Übernickel et al., 2021a), we calculated the daily precipitation in mm. The sediment redistribution
recorded immediately and within five scans before and after a rainfall event is defined to be the result of the
rainfall event. The five-scan buffer is necessary as the climate stations are located up to a 15 km distance from
the cameras (Fig. 1). We calculated the rainfall-caused sediment redistribution within (i) areas affected by the
burrowing animal (i.e., entrance, mound and burrow roof) and (ii) within areas not affected by the burrowing
animal. To estimate the sediment volume which accumulated within the entrance, we also calculated the
volume of redistributed sediment solely (iii) within the mound and burrow roof. From this volume, we subtracted
the volume redistributed within all affected areas. We did not directly calculate the accumulation rate within the
entrance as the cameras did not provide data on the complete underground burrow structure.

**3.8 Hillslope-wide upscaling of redistributed sediment**



Hillslope-wide upscaling of the results generated in this study was performed by utilising a previous
estimation of vertebrate burrow density (Grigusova et al., 2021). In this study, the density of burrows created
by vertebrate burrowing animals, which we interpreted as vertebrate burrow density (number of burrows per
100 m²), was measured in situ within eighty 100 m² plots and then upscaled to the same hillslopes on which
the cameras were located by applying machine-learning methods, using the UAV-data as predictors. Hence,
the modelled burrows in the previous study were in fact areas affected by burrowing animals in this study.
From the camera data, we calculated the average cumulative volume of redistributed sediment after
a period of one year within affected ($Vol_{affected}$ ($cm^3$ $cm^{-2}$ $year^{-1}$)) and non-affected ($Vol_{not\ affected}$ ($cm^3$ $cm^{-2}$ $year^{-1}$)) areas and the average sediment volume redistributed (excavated) by the animal ($Vol_{exc}$ ($cm^3$ $cm^{-2}$ $year^{-1}$)),
separately for each site. Additionally, we estimated the volume of sediment that was additionally redistributed
during rainfall events due to the presence of the burrow ($Vol_{add}$ ($cm^3$ $cm^{-2}$ $year^{-1}$)). $Vol_{add}$ was calculated as the
difference in the redistributed sediment volume between affected and non-affected areas according to Eq. (7).
$$Vol_{add} = \frac{(M_{affected} - M_{unaffected})}{7} * 12 \qquad , \qquad\qquad (7)$$
We then upscaled the $Vol_{affected}$ ($cm^3$ $cm^{-2}$ $year^{-1}$), $Vol_{exc}$ ($cm^3$ $cm^{-2}$ $year^{-1}$)) and $Vol_{add}$ ($cm^3$ $cm^{-2}$ $year^{-1}$)) to the hillslope using the same approach: First, we calculated the average volume of the redistributed
sediment per burrow ($Vol_{per\ burrow}$ [$cm^3$ $burrow^{-1}$ $year^{-1}$]):
$$Vol_{per\ burrow} = \frac{(Area_{burrow} * Vol)}{7} * 12 \qquad\qquad (8)$$
In Eq. (8), $Area_{burrow}$ ($cm^2$) is the average size of the burrows that are monitored by the cameras; Vol is $Vol_{affected}$
($cm^3$ $cm^{-2}$ $year^{-1}$), $Vol_{exc}$ ($cm^3$ $cm^{-2}$ $year^{-1}$) or $Vol_{add}$ ($cm^3$ $cm^{-2}$ $year^{-1}$). Using the hillslope-wide predicted
vertebrate burrow densities ($Dens_{burrow}$ (number of burrows 100 $m^{-2}$)) from Grigusova et al. 2021, we estimated
the volume of redistributed sediment for each pixel of the raster layers ($Vol_{per\ pixel}$ ($cm^3$ $m^{-2}$ $year^{-1}$)) according
to Eq. (9):
$$Vol_{per\ pixel} = \frac{Vol_{per\ burrow} * Dens_{burrow}}{7} * 12 \qquad\qquad (9)$$
The average hillslope-wide volume of redistributed sediment ($Vol_{hillslope\text{-}wide}$ ($m^3$ $ha^{-1}$ $year^{-1}$)) was then
estimated as follows:
$$Vol_{hillslope-wide} = \frac{\sum_{1}^{m} Vol_{per\ pixel}}{7} * 12 * 0.001 \qquad , \qquad\qquad (10)$$
In Eq (10), m is the number of pixels.

**4. Results**
**4.1 Camera accuracy**
The accuracy between the measured extracted sediment volume and sediment volume calculated
from the camera scans was very high (MAE = 0.023 $cm^3$ $cm^{-2}$, $R^2$ = 0.77, Fig. 4). The accuracy between the
calculated and measured extracted sediment was higher when the two scans taken before as well as after the
extraction of the sediment were averaged and the sediment volume was estimated using these averaged
scans. When calculating the redistributed sediment from solely one scan before and after extraction, the
accuracy slightly decreased (MAE = 0.081 $cm^3$ $cm^{-2}$, $R^2$ = 0.64). The cameras tended to overestimate the
volume of redistributed sediment.





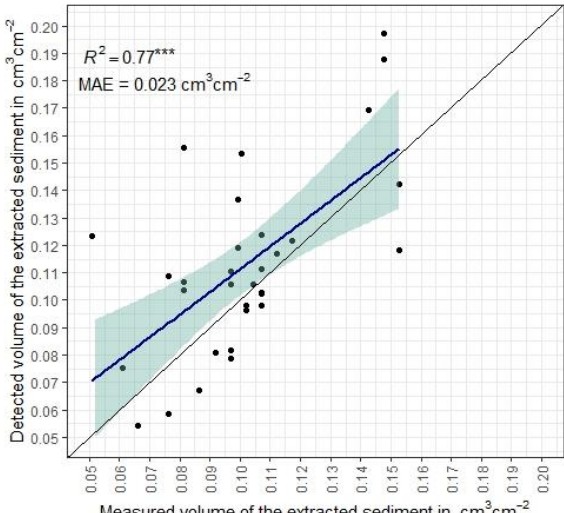

**Figure 4.** Estimation of Time-of-Flight camera accuracy based on averaging two surface scans before and after the sediment extraction under controlled conditions. The *x*-axis shows the exact sediment volume measured with a cup. The *y*-axis represents the volume of the sediment calculated from the camera scans (according to Equation (4)). The blue line is the linear regression calculated from the measured and detected volume. The green shadow shows the confidence interval of 95% for the linear regression slope. ***$p \leq 0.001$. MAE is the mean absolute error, and $R^2$ the coefficient of determination.

### 4.2 Data quantity and quality

Six out of eight custom-tailored cameras collected data over the seven-month period (Table A2). One camera collected data for a period of three months and one camera stopped working a few days after installation. The quantity of usable point clouds taken at 1 a.m., 5 a.m. and 10 p.m. was higher than of point clouds taken at 8 a.m. Approximately 20% of points was removed from the point clouds before final analysis due to the high scattering at the point cloud corners. After data filtering (see Section 3.2.), 1326 scans were usable and for 86% of the days, at least one usable scan was available. The usable scans were distributed continuously within the monitoring period.

In LC, the areas affected by the burrowing animal always consisted of an entrance, mound and burrow roof. In PdA, there was no burrow roof on the upper hillslopes. Burrows without a burrow roof were located on shallower parts of the hillslopes (up to an inclination of 5°), and the angle of the burrow entrance to the ground was ~90°. Burrows with a burrow roof were located on steeper parts of the hillslopes (with an inclination above 5°), and the angle of the burrow entrance to the ground was ~45°.

### 4.3 Mass balance of redistributed sediment

The cameras detected (i) sediment redistribution directly following rainfall events and (ii) due to the burrowing activity in times without rainfall (Fig. 5 and 6). In all cases, areas affected by burrowing activity (entrance, burrow roof and mound) exhibited higher sediment redistribution rates than areas not affected by burrowing. In addition, the volume of redistributed sediment by animal activity was higher after a rainfall event



occurred. After rainfall events, sediment eroded especially at the boundaries of the burrow roof and mound.
Sediment eroded from the burrow roof accumulated within the burrow entrance or was redistributed to the
mound (Fig. 7). We can also identify detected sediment accumulation on the upper parts of the burrow roof.
After the burrowing activity, sediment was excavated from the entrance and cameras detected accumulation
on the mound and within the burrow roof. The sediment accumulation and erosion within the area not affected
by the animal was evenly distributed around the burrow (Fig. 7).

In the following, the dynamics are exemplarily explained for four cameras. Animal burrowing activity
was detected seven times by the camera LC NU (Fig. 5a, A2a and A3a) during the monitoring period, by an
increase in sediment volume in the area delineated as mound. Simultaneously, the burrow entrance showed
signs of modification and sediment accumulation, but these changes were less clear. Overall, the volume of
the excavated soil varied. From April until June, up to 0.5 $cm^3$ $cm^{-2}$ of sediment was excavated by the animal
and accumulated on the mound. From June until September, animal burrowing activity was detected at four
time slots (5 June 2019, 9 June 2019, 1 July 2019 and 18 August 2019) and sediment volume of up to 2 $cm^3$
$cm^{-2}$ accumulated each time on the mound, burrow roof and within the entrance. During the rainfall events of
up to 20 mm $day^{-1}$ on 16 June 2019, 27 mm $day^{-1}$ on 29 June 2019 and 7 mm $day^{-1}$ on 13 July 2019, sediment
volume of up to 4 $cm^3$ $cm^{-2}$ eroded, especially from the burrow roof and the mound while a sediment volume
of up to 1 $cm^3$ $cm^{-2}$ accumulated within the entrance during each rainfall event.

Camera LC-SL (Fig. 5b and A3b) showed burrowing activities eight times and sediment volumes of up
to 3 $cm^3$ $cm^{-2}$ accumulated within the entrance and burrow roof. The camera detected sediment erosion of up
to 2 $cm^3$ $cm^{-2}$ after a rainfall event of 27 mm $day^{-1}$ on 27 July 2019. On the south-upper hillslope, the camera
detected animal burrowing activity six times, with a sediment accumulation of up to 3 $cm^3$ $cm^{-2}$ (Fig. A2 and
A3).

In contrast, camera PdA-NU pointed to animal burrowing activity up to 15 times where up to 1 $cm^3$ $cm^{-}$
$^{2}$ of sediment volume was redistributed from the entrance to the mound (Fig. 6a, A2e and A3e). At the end of
June on 27 June 2019, a rainfall event of 1.5 mm $day^{-1}$ occurred and up to 2 $cm^3$ $cm^{-2}$ of sediment eroded
from the burrow roof and accumulated within the burrow entrance. We observed increased sediment
redistribution by the animal after the rainfall events.

Camera PdA-SL evenly revealed animal burrowing activity up to 15 times (Fig. 6b, A2f and A3f). The
burrowing had a strong effect on the sediment redistribution. The rainfall event of 1.5 mm $day^{-1}$ on 27 June
2019 did not cause any detectable surface change.

Earth **Surface**
**Dynamics**
Discussions



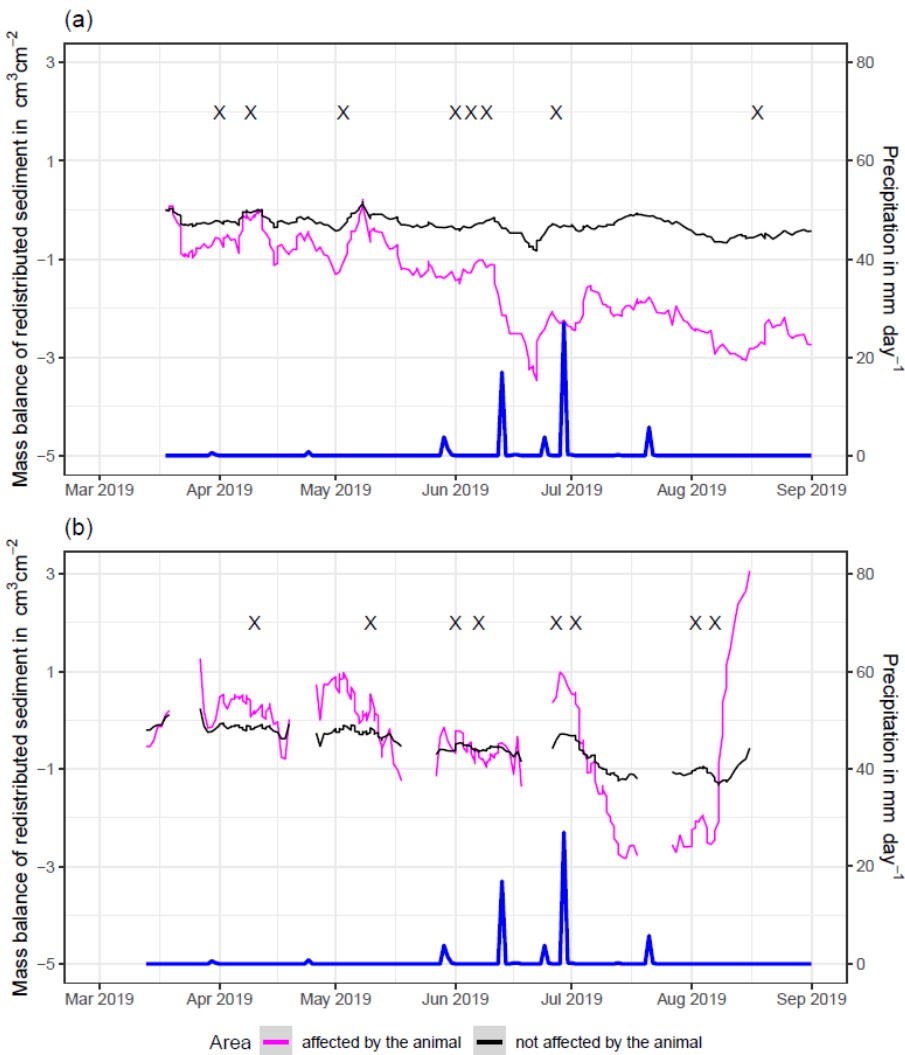

**Figure 5.** Examples of mass balance of redistributed sediment in mediterranean La Campana for areas affected and not affected by burrowing animals: (**a**) The record of the camera on the upper north-facing hillslope showed that larger rainfall events cause a negative sediment balance (sediment loss), followed by a phase of positive sediment mass balance after approximately 3 days due to the fact of sediment excavation; (**b**) The record of the camera on the lower south-facing hillslope showed a similar pattern to the camera on the upper north-facing hillslope, but the phase of positive mass balance was delayed in comparison. The blue line is the daily precipitation in mm day$^{-1}$, and "X" marks the days at which animal burrowing activity was detected. Mass balances for all cameras are displayed in Fig. A2 and A3.



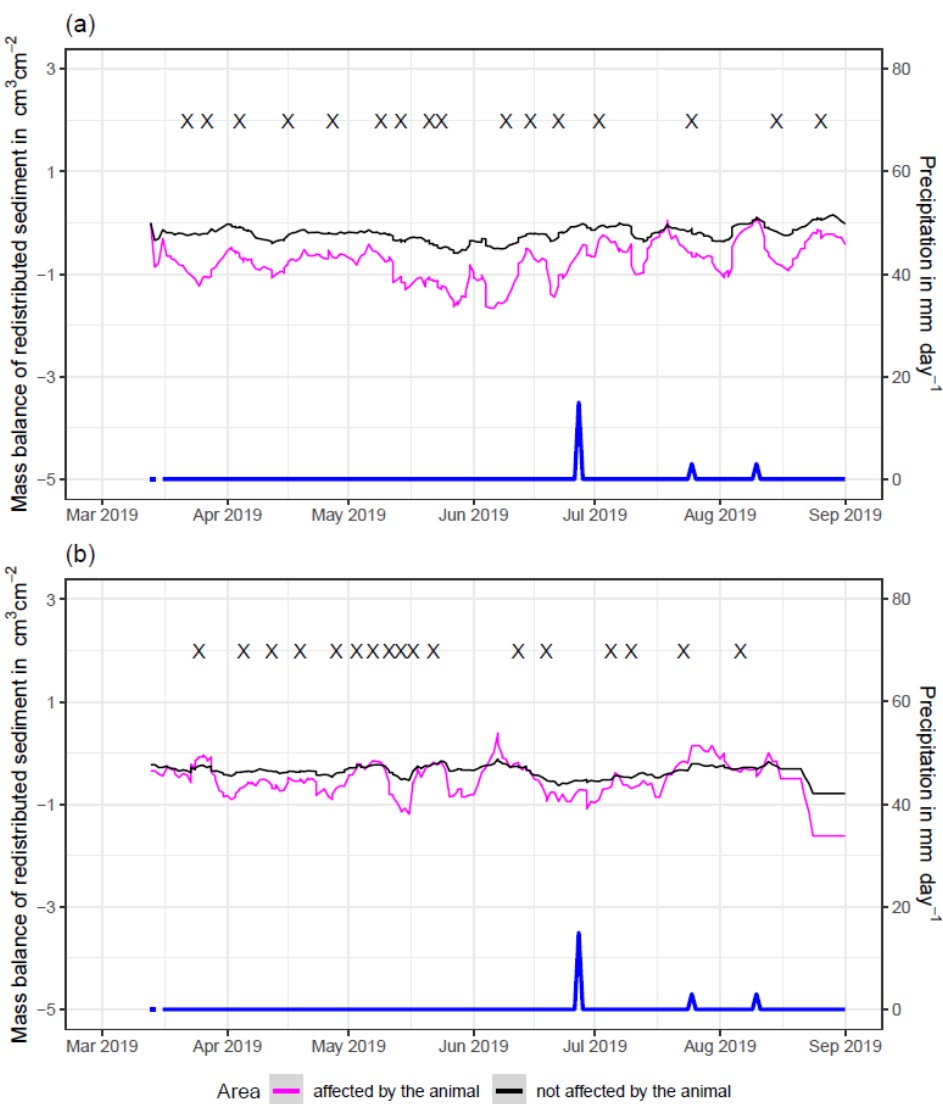

**Figure 6.** Examples of mass balance of redistributed sediment in the arid Pan de Azúcar for areas affected and not affected by burrowing animals: (**a**) The record of the camera on the upper north-facing hillslope showed that animal sediment excavation had a larger impact on the mass balance than rainfall events; (**b**) The record of the camera on the lower south-facing hillslope showed a similar pattern to the camera on the upper north-facing hillslope. The blue line is the daily precipitation in mm day$^{-1}$, and "X" marks the days at which animal burrowing activity was detected. Mass balances for all cameras are displayed in Fig. A2 and A3.



Earth **Surface**
**Dynamics**
Discussions



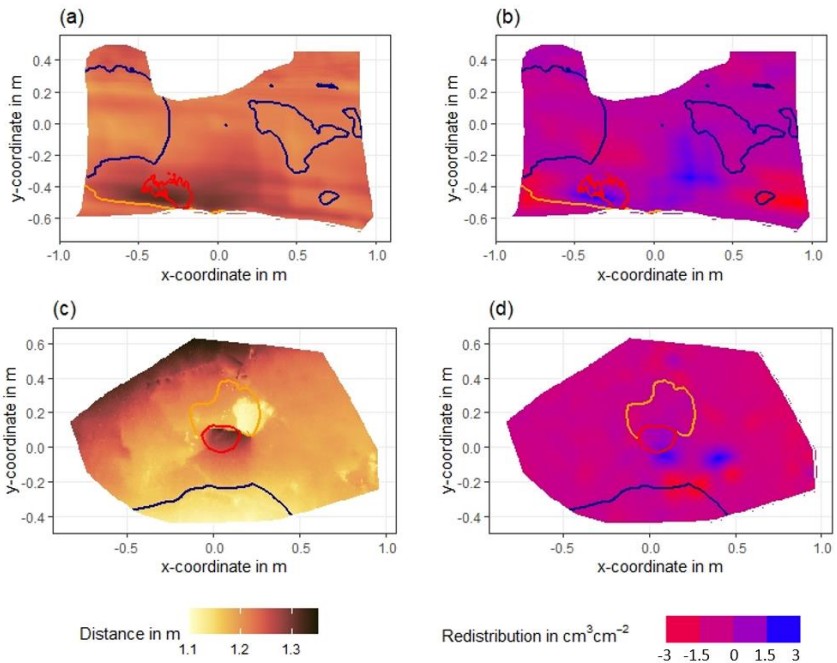

**Figure 7.** Examples of surface scans showing the digital surface model (DSM) before a rainfall event (**a**, **c**) at two camera locations in La Campana, and the calculated volume of redistributed sediment (**b**, **d**) after the rainfall event: (**a**) DSM of a scan from the camera on the upper north-facing hillslope in La Campana; (**b**) Detected sediment redistribution (cm³ cm⁻²) on the upper north-facing hillslope in La Campana after a rainfall event of 17.2 mm day⁻¹; (**c**) DSM of a scan from the camera on the upper south-facing hillslope in La Campana; (**d**) Detected sediment redistribution (cm³ cm⁻²) on the upper south-facing hillslope after a rainfall event of 17.2 mm day⁻¹. Red is the outline of the burrow entrance. Green is the outline of mound. Orange is the outline of the burrow roof. The area which is not outlined is the area not directly affected by animal burrowing activity. Redistribution is the volume of the redistributed sediment, either accumulated (positive value) or eroded (negative value) per cm³ cm⁻². After the rainfall events, sediment mostly accumulated within the burrow entrance or near mounds and eroded from burrow roofs and mounds.

### 4.4 Cumulative volume of redistributed sediment

The analysis of cumulative volume of the redistributed sediment caused by burrowing animal activity and rainfall over the monitored period of seven months for all eight cameras showed a heterogeneous pattern (Fig. 8 and A4, Tables A3 and A4).

In LC, the cumulative volume of the sediment excavated by the animal within the burrow roof and mound increased continuously (Fig. 8). Especially between the rainfall events from June until August, a cumulative volume of on average 6.5 cm³ cm⁻² was excavated by the animal. We calculated that, on average, 8.53 cm³ cm⁻² cumulatively eroded from the burrow roof and mound; while 2.44 cm³ cm⁻² sediment volume accumulated within the entrance (Fig. 8). These results indicate that 28% of sediment eroding from the burrow roof accumulated within the entrance, while over 62% of sediment eroded downhill. Averaged over all camera



scans, 338% more sediment was redistributed by rain within the affected area compared to the non-affected
area (Fig. 9).
In PdA, cameras continuously detected animal burrowing activity and excavation of the sediment (Fig.
A4). The volume of the detected excavated sediment increased steadily within all cameras. The cumulative
sediment accumulation surpasses the sediment eroded due to the rainfall. The volume of the sediment eroded
within the affected areas was 40% higher than within the non-affected areas (Fig. 9). The results show that
approximately 50% of the eroded sediment accumulated within the entrance.

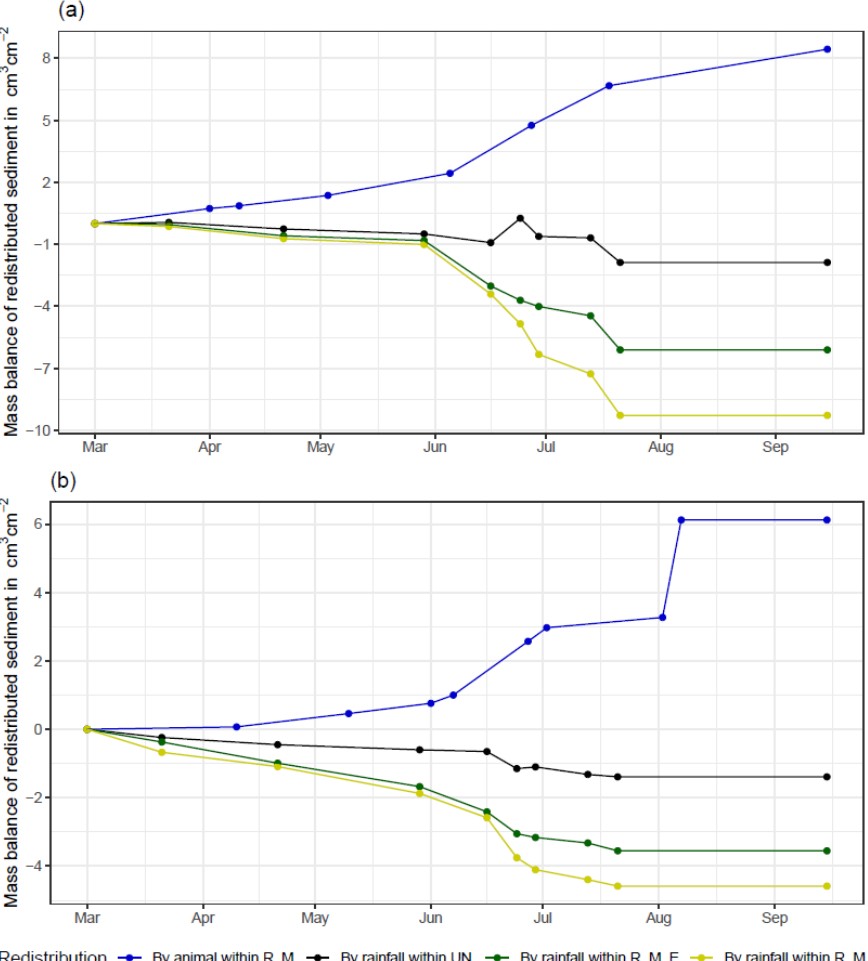

Redistribution   ● By animal within R, M   ● By rainfall within UN   ● By rainfall within R, M, E   ● By rainfall within R, M
**Figure 8.** Examples of the cumulative volume of redistributed sediment within affected and non-affected areas
caused by animal burrowing activity or rainfall in mediterranean La Campana: (**a**) Upper north-facing hillslope;
(**b**) Lower south-facing hillslope. Positive values indicate sediment accumulation. Negative values indicate
sediment erosion. E is the burrow entrance; M is the mound; R is burrow roof; UN is the area not directly
affected by the animal burrowing activity. Cumulative volumes for all cameras are in Fig. A4.




Earth **Surface** Dynamics
Discussions

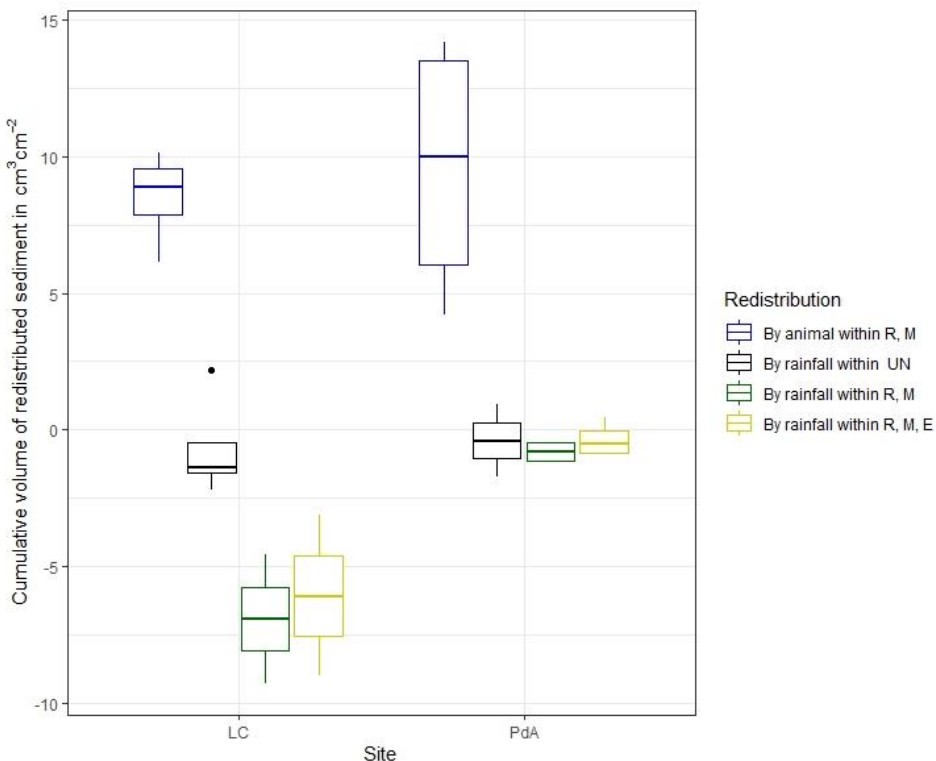

**Figure 9.** Cumulative volume of the redistributed sediment for all cameras. Positive values indicate sediment accumulation. Negative values indicate sediment erosion. Whiskers indicate the median of sediment redistribution. E is the burrow entrance; M the mound; R is the burrow roof; UN is area not affected by the animal burrowing activity; LC stands for National Park La Campana in the mediterranean climate zone; PdA stands for National Park PdA in the arid climate zone.

**4.5 Hillslope wide excavation and redistribution**

Grigusova et al. (2021) showed that the density of vertebrate burrows was between 6 and 12 per 100 $m^2$ in LC and between 0 and 12 per 100 $m^{-2}$ in Pan de Azúcar. The volume of the sediment excavated by the animal and redistributed during rainfall events varied between sites and across the hillslopes (Fig. 10, A5 and A6; Tables A3 and A4).

The volume of the sediment excavated by the animal per burrow was lower in LC (1226.61 $cm^3$ $burrow^{-1}$ $year^{-1}$) than in PdA (1498.66 $cm^3$ $burrow^{-1}$ $year^{-1}$) (Fig. 9, Table 1). However, on the hillslope scale, a higher total area-wide volume of excavations was calculated for LC compared to PdA (0.67 $m^3$ $ha^{-1}$ $year^{+1}$ vs. 0.18 $m^3$ $ha^{-1}$ $year^{-1}$), due to the higher burrow density in LC.

The volume of the sediment redistributed within the area affected by burrowing activity during rainfall events on the hillslope scale was higher in LC (–0.48 $m^3$ $ha^{-1}$ $year^{-1}$) than in PdA (–0.05 $m^3$ $ha^{-1}$ $year^{-1}$) (Table 1). The volume additionally redistributed sediment due to the presence of the burrows was well higher in LC than in PdA (Fig. 10, A5 and A6).

The hillslope-wide redistribution rates increased with burrow density, which, as stated in Grigusova et al. (2021), largely depends on vegetation distribution and topography. In LC, more sediment was excavated in





parts of the hillslope with a vegetation cover of over 50% m$^{-2}$ (~0.60 m$^3$ ha$^{-1}$ year$^{-1}$) than in non-vegetated
parts of the hillslope (~0.2 m$^3$ ha$^{-1}$ year$^{-1}$). However, dense vegetation covers over 80% m$^{-2}$ reduced volume
of redistributed sediment due to the animals' burrowing activity (Fig. A5, A7c and A7d). More sediment was
redistributed in the middle and upper parts of the hillslope (Fig. A5, A7a and A7b). In PdA, the volume of
sediment redistributed by the burrowing animals increased with vegetation cover and elevation (Fig. A6 and
A8).

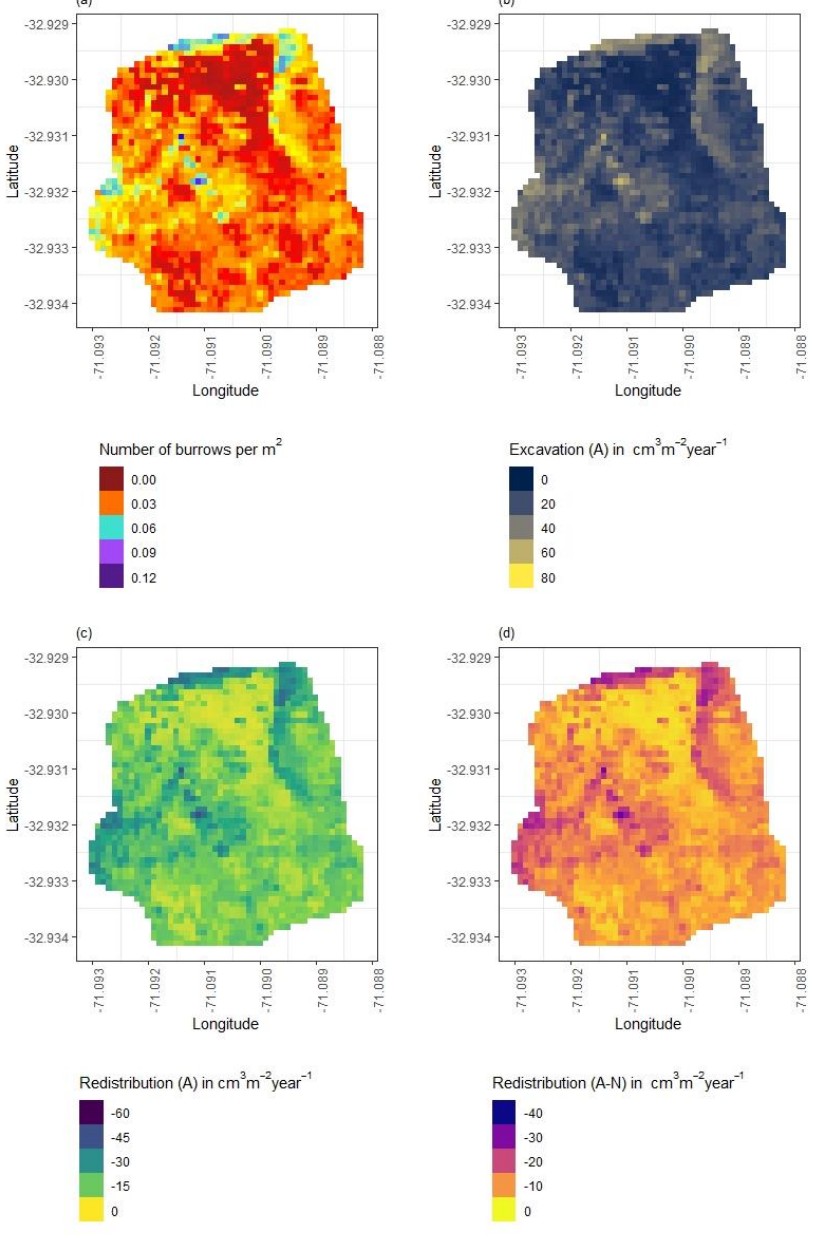






**Figure 10.** Example of the hillslope-wide volume of redistributed sediment for a time period of one year on the
south-facing hillslope in La Campana: (a) Density of burrows as estimated by Grigusova et al. (2021); (b)
Volume of the sediment excavated by the animals; (c) Volume of the sediment redistributed during rainfall
events within affected areas; (d) Volume of additionally redistributed sediment during rainfall events due to the
presence of the burrows. The values were calculated per burrow as stated in Section 3.7. by subtracting the
sediment volume redistributed within animal-affected areas from the sediment volume redistributed within non-
affected area and then upscaled. The letters in brackets indicate if the upscaling was conducted using data
from affected or non-affected areas by burrowing animals. "A" stands for affected area. By "A-N", the
redistribution calculated from non-affected areas was subtracted from the redistribution calculated within
affected areas to obtain the additional volume of redistributed sediment due to the burrows' presence.

**Table 1.** Summary of the volume of redistributed sediment according to site, area, and disturbance type. $Vol_{exc}$
describes the volume of the sediment excavated by the animals. $Vol_{affected}$ describes the volume of the sediment
redistributed during rainfall events within affected areas. $Vol_{add}$ describes the volume of additionally
redistributed sediment during rainfall events due to the presence of the burrows. The values were calculated
per burrow, as stated in Section 3.7., by subtracting the sediment volume redistributed within animal-affected
areas from the sediment volume redistributed within non-affected areas and then upscaled.

| Variable | | Volume of redistributed sediment | |
|---|---|---|---|
| **Disturbance** | Area | LC | Pan de Azúcar |
| **$Vol_{exc}$** | Affected area | 14.62 cm$^3$ cm$^{-2}$ year$^{-1}$ | 16.41 cm$^3$ cm$^{-2}$ year$^{-1}$ |
| | Per burrow | 1226.61 cm$^3$ burrow$^{-1}$ year$^{-1}$ | 1498.66 cm$^3$ burrow$^{-1}$ year$^{-1}$ |
| | Hillslope-wide | 0.67 m$^3$ ha$^{-1}$ year$^{-1}$ | 0.18 m$^3$ ha$^{-1}$ year$^{-1}$ |
| **$Vol_{affected}$** | Affected area | −10.44 cm$^3$ cm$^{-2}$ year$^{-1}$ | −1.41 cm$^3$ cm$^{-2}$ year$^{-1}$ |
| | Per burrow | −876.38 cm$^3$ burrow$^{-1}$ year$^{-1}$ | −126.36 cm$^3$ burrow$^{-1}$ year$^{-1}$ |
| | Hillslope-wide | −0.48 m$^3$ ha$^{-1}$ year$^{-1}$ | −0.05 m$^3$ ha$^{-1}$ year$^{-1}$ |
| **$Vol_{add}$** | Affected area | −7.37 cm$^3$ cm$^{-2}$ year$^{-1}$ | −1.18 cm$^3$ cm$^{-2}$ year$^{-1}$ |
| | Per burrow | −619.2 cm$^3$ burrow$^{-1}$ year$^{-1}$ | −48.36 cm$^3$ burrow$^{-1}$ year$^{-1}$ |
| | Hillslope-wide | -0.34 m$^3$ ha$^{-1}$ year$^{-1}$ | -0.02 m$^3$ ha$^{-1}$ year$^{-1}$ |



**5. Discussion**
Our results showed that the new ToF device is a suitable tool for high-resolution, autonomous
monitoring of surface changes, applicable also in remote areas. The ability of a continuous observation of
sediment redistribution over a longer time during our study provided new insights into the importance of
burrowing animals for sediment redistribution. Our research reveals that the presence of vertebrate burrows
increases hillslope sediment redistribution rates much more than previously assumed (up to 208%). We
showed that the quantity of animal-related sediment redistribution, however, varied on rainfall occurrence, with
an increase in sediment redistribution between 40% in the arid research area and 338% percent in the
mediterranean research area.

**5.1 Suitability of the ToF method for surface monitoring**





The here proposed monitoring technique enables an automatic monitoring of surface changes on a
microtopographic scale, and its measurement continuity allows for the analysis of ongoing
biogeomorphological processes in high temporal resolution.
Our ToF device stands in contrast to earlier studies that used laser scanning technology to monitor
microtopographic changes, with regard to the costs, measurement frequency and sampling autonomy (Table
A5).
Previous authors mainly applied expensive laser scanning for the estimation of sediment redistribution,
and the research sites had to be personally revisited for each of the measurements (Nasermoaddeli and
Pasche, 2008; Eltner et al., 2016b; Eltner et al., 2016a; Hänsel et al., 2016). In contrast to other laser scanning
methods (Table A5), our approach is instrumentally much more cost-effective and, thus, easier to apply and
more available also for smaller case studies with multiple objects to be observed at the same time. The cost
for the laser scanners and needed equipment used in previous studies varied between USD 4500  and up to
USD 240,000  (Nasermoaddeli and Pasche, 2008; Morris et al., 2011) (Table A5). In comparison, the cost of
our ToF system comprises only USD 900, which is a 5–240 times lower price.
In terms of data quality, our ToF device is more precise or comparable to those employed in other
studies. The accuracy of the camera ($R^2$ = 0.77) was in the range of previous studies ($R^2$ = 0.26–0.83 (Eitel et
al., 2011), Table A5). The horizontal point spacing of our cameras was 0.32 cm, and the maximum number of
points per cm$^2$ was 8.5. These values are similar to previous studies in which the used devices had a horizontal
point spacing in the range of 0.25–0.57 cm (Kaiser et al., 2014; Nasermoaddeli and Pasche, 2008)) (Table
A5), and the maximum number of points per cm$^2$ in a range of 1 point–25 points cm$^{-2}$ (Eitel et al., 2011; Longoni
et al., 2016) (Table A5).
Our cameras tended to slightly overestimate or underestimate the volume of redistributed sediment.
This error occurs when the pulse reflects from several vertical objects such as walls or, in our case, branches
or stones and then enters the camera sensor. This phenomenon was also observed in previous studies
applying laser scanners and is inevitable if the goal is to study surface changes under natural field conditions
(Kukko and Hyyppä, 2009; Ashcroft et al., 2014). During operation of the cameras, we learnt that our newly
developed instruments are particularly capable of delivering usable scans at night. This is likely due to the
strong scattered sunlight reaching the camera sensor during the day, blurring the data (Li, 2014). Thus, we
recommend focusing on nocturnal operation to prevent light contamination from the surroundings.
We could thus prove that the ToF cameras are a suitable and cost-effective method for a continuous
monitoring of sediment redistribution at a microtopographic scale without the need of expensive laser scanning
campaigns.

**5.2. Sediment Redistribution**
Our research reveals that the presence of vertebrate burrows generally increases hillslope sediment
redistribution. We show, however, that the ratio between the sediment redistribution caused by rainfall in the
areas affected and not affected by burrowing animals varies between the climate zones. The sediment
redistribution in the affected areas was 40% higher in the arid research site, and in the mediterranean research
site, it was 338% higher when compared to the areas not affected by burrowing animals (Fig. 9, Tables A3 and
A4).
By monitoring microtopographical changes in a high spatio-temporal resolution, we found that the
occurrence of larger rainfall events played a two-fold, accelerating role in influencing sediment redistribution





(Fig. A3 and A4). Firstly, rainfall-runoff eroded burrow material causing increased sediment loss. Ultimately,
after the rainfall, the cameras detected animal burrowing activity. The rainfall triggered the burrowing activity
which was likely related to a lower burrowing resistance of the soil due to the increased soil moisture (Rutin,
1996; Romañach et al., 2005; Herbst and Bennett, 2006). This double feedback led to frequently occurring but
small redistribution rates. However, cumulatively, this mechanism increased downhill sediment fluxes. Previous
studies most likely missed this low magnitude but frequent surface processes and, thus, did not estimate the
full volume of redistributed sediment due to the fact of their sporadically taken measurements. To quantify all
occurred sediment redistribution processes, a continuous frequent surface monitoring, like the here presented,
is needed.
In contrast to our result, the maximum increase in the sediment volume redistributed during rainfall
events measured in the areas affected by burrowing animals when compared to not affected areas was 208%
( (Imeson and Kwaad, 1976) Table A6). Our results, however, indicate an increase of up to 338% (Table 9).
This means, that the contribution of animals' (vertebrates') burrowing activity in the mediterranean climate
appear larger than previously observed by using field methods such as erosion pins or splash traps (–3 –
208%, (Imeson and Kwaad, 1976; Hazelhoff et al., 1981; Black and Montgomery, 1991) , Table A6). In contrast,
in arid PdA, our study found a much smaller increase (40%, Fig. 9) in the sediment volume redistributed during
rainfall events measured in the areas affected by burrowing animals when compared to not affected areas.
This is lower than previously estimated (125%, (Black and Montgomery, 1991), Table A6). However, solely
one rainfall event above 0.2 mm day$^{-1}$ occurred during our monitoring period. Hence, we conclude that the
contribution of burrowing activity of animals to hillslope sediment transport is much larger in areas with frequent
rainfall events than previously thought, while it has been realistically estimated by previous studies for areas
with rare rainfall events (Table A6).
Overall, our study revealed a strong impact of animal excavation processes on sediment redistribution
in the mediterranean climate zone (0.67 m$^3$ ha$^{-1}$ year$^{-1}$), which were more in a range of excavated volume
observed in previous studies by bears and porcupines (0.49 m$^3$ ha$^{-1}$ year$^{-1}$, (Hall et al., 1999), Table A8) than
rodents (0.02 m$^3$ ha$^{-1}$ year$^{-1}$, (Hall et al., 1999)). The estimated sediment excavation in the arid climate zone
(0.18 m$^3$ ha$^{-1}$ year$^{-1}$, Fig. A4, Table A8) was in the order of magnitude of previous studies (0.05–0.2 m$^3$ ha$^{-1}$
year$^{-1}$, (Black and Montgomery, 1991; Yoo et al., 2005), Table A8). Our results thus suggest that animal
burrowing activity is an important part of the environmental mechanisms leading to increased sediment fluxes
in wetter (as a consequence of animal-triggered excavation and rainfall-triggered erosion) and drier (as a
consequence of animal-triggered excavation) regions (Fig. 11).

Earth **Surface**
**Dynamics**
Discussions

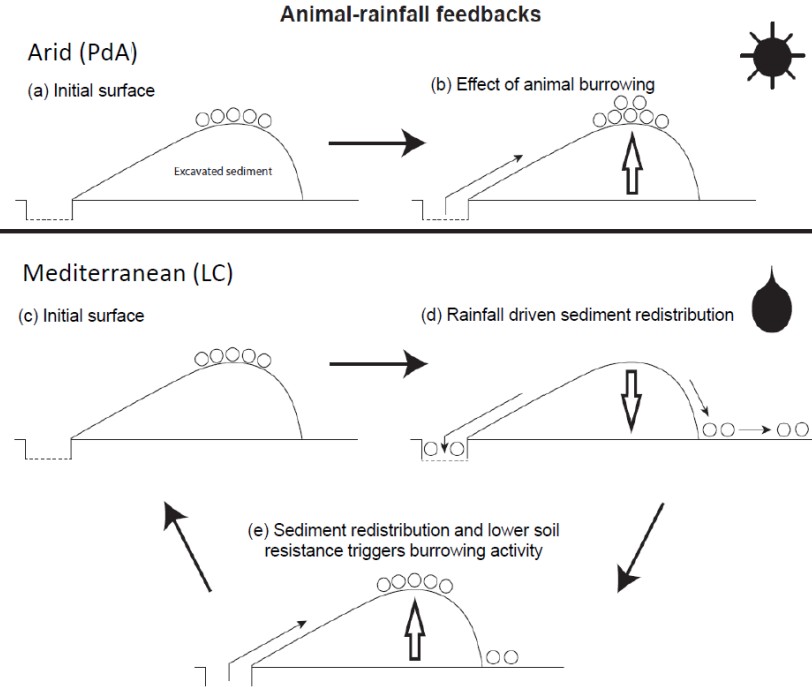

**Figure 11.** Scheme of the animal-driven and rainfall-driven sediment redistribution processes in both climate zones: (**a**) Describes the initial surface of the burrow before a start of a redistribution process and (**b**) the animal excavation process in the arid zone. Here, due to sporadically occurring rainfall events, sediment redistribution is mostly controlled by the animal burrowing activity; (**c**) Describes the initial burrow surface in the mediterranean climate zone, (**d**) the process of sediment redistribution during a rainfall event and (**e**) the following animal burrowing activity. The burrowing is triggered by decreased soil resistance due to the increased soil moisture after rainfall as well as by sediment accumulation within the burrow's entrance. The burrowing activity leads to a new supply of sediment being excavated to the surface. In the mediterranean climate zone, the sediment redistribution is controlled by both animal burrowing activity and rainfall. The alternating excavation and erosion process ultimately lead to an increase in redistribution rates.

Magnitudes of sediment volume redistributed within areas affected by burrowing animals similar to our results were previously obtained solely in studies applying rainfall simulator. These studies estimated an increase in the volume of sediment redistributed during rainfall events, measured in the areas affected by burrowing animals when compared to not affected areas, to be between 205% and 473% (Li et al., 2018; Chen et al., 2021), Table A6). However, a rainfall simulator can only provide data on surface processes within a plot of a few m² in size and under ideal laboratory conditions while ignoring the uphill microtopography, vegetation cover and distribution (Iserloh et al., 2013) which were shown to reduce erosion rates. More importantly, the rainfall intensity on hillslopes decreases with (i) the angle of incidence of the rain, (ii) the inclination of the surface and iii) the relative orientation of the sloping surface to the rain vector (Sharon, 1980). When simulating a rainfall event with the same rainfall volume as in the field, the rain is induced directly over the treated surface and has thus a higher velocity which leads to an increased splash erosion than under natural conditions (Iserloh



et al., 2013). We thus propose that the rainfall experiments overestimate the erosion rate while the correct
erosion rate can be measured solely under field conditions.

Cumulative sediment redistribution within burrow roof, mound and entrance was, on average, 28%

lower than cumulative sediment redistribution only within the mound and the burrow roof (Fig. 9 and A4). These
results suggest that 28% of the eroded sediment from animal mounds and burrow roofs is re-accumulated
within the burrow entrance during rainfall-runoff events, and the remaining 62% is incorporated into overall
hillslope sediment flux. Our numbers contrast with previous studies, which quantified that about 58% of the
sediment excavated by animals will accumulate back in the burrow entrance and only 42% is incorporated to
downhill sediment flux (Andersen, 1987; Reichman and Seabloom, 2002). Hence, our results indicate not only
higher redistribution rates within areas affected by the burrowing animals but also point to much higher supply
of sediment to the downhill sediment flux as previously thought.

On the hillslope scale, the contrast between our estimated volume of redistributed sediment during

rainfall events within areas affected by burrowing animals (from $-0.67$ until $-1.18$ cm$^3$ cm$^{-2}$ year$^{-1}$) and the
previous studies was even higher than on the burrow scale (from $-0.183$ until $-1.56$ cm$^3$ cm$^{-2}$ year$^{-1}$, (Imeson
and Kwaad, 1976; Li et al., 2018), Table A7). This was well pronounced when estimating the hillslope-wide
volume of the sediment excavated by the animals (from 0.18 cm$^3$ cm$^{-2}$ year$^{-1}$ until 0.67 cm$^3$ cm$^{-2}$ year$^{-1}$
according to our study, and from 0.05 cm$^3$ cm$^{-2}$ year$^{-1}$ until 0.49 cm$^3$ cm$^{-2}$ year$^{-1}$ according to previous studies,
(Black and Montgomery, 1991; Hall et al., 1999), Table A8). The previous studies estimated the area-wide
excavated soil volume solely once or twice a year. We propose that these measurements only describe the
current burrow distribution, however, cannot consider the continuous excavation and erosion dynamics.

Our study offers new insights previously undescribed in literature. Our cost-effective ToF device

provides data on surface changes in a high spatio-temporal resolution. The high temporal resolution could
unravel ongoing low magnitude but frequent excavation and erosion processes. High spatial resolution enabled
us to estimate the exact volume of sediment fluxes from the burrows downhill. Our results indicate that the
contribution of burrowing animals on the burrow as well as on the hillslope scale was much higher than
previously assumed. In our future research, we intend to include our findings into long-term soil erosion models
that rely on soil processes but do not yet include animal-induced surface processes on microtopographical
scales in their algorithms.

**6. Conclusion**

Our study provides new insights on the impacts of burrowing animals on hillslope sediment fluxes. The

continuous high-resolution monitoring enables to study the surface processes in detail and the high temporal
data availability revealed higher redistribution rates within areas affected by burrowing animals than previously
assumed. We discovered an alteration between sediment excavation by the animal and sediment erosion
during rainfall events which unveil a continuous sediment contribution of burrowing animals to hillslope
sediment flux. Although we concentrated on the impacts of burrowing animals on sediment redistribution, the
applicability of the cameras is not limited to our research topic. Other possible applications could, for example,
be a study of surface roughness, impacts of dead wood on erosion, biomass changes throughout the year or
decomposition processes.

**Funding:** This study was funded by the German Research Foundation, DFG [grant numbers
BE1780/52-1, LA3521/1-1, FA 925/12-1, BR 1293-18-1], and is part of the DFG Priority Programme





SPP 1803: EarthShape: Earth Surface Shaping by Biota, sub-project "Effects of bioturbation on rates
of vertical and horizontal sediment and nutrient fluxes".
**Institutional Review Board Statement:** Not applicable.
**Informed Consent Statement:** Not applicable.
**Acknowledgments:** We thank CONAF for the kind support provided during our field campaign.
**Competing interests:** There is no conflict of interest.
**Author contribution:** JB, AL and SA planned the campaign; PG and SA performed the measurements; PG
analysed the data and wrote the manuscript draft; AL, JB, NF, RB, KÜ, LP, CR, DK and PP reviewed and edited
the manuscript.
**Code/Data availability:** Code and all raw data can be provided by the corresponding author upon request.


**Appendices**
**Table A1.** List of abbreviations

| α [°] | **Tilt angle of the camera** |
|---|---|
| b [°] | Surface inclination |
| Ω | Threshold value for the scan scattering error |
| A | Affected area |
| Affected area | Area directly affected by the burrowing animal |
| $Area_{burrow}$ | mean in the field measured size of the burrows which are monitored |
| Area | total surface area monitored by the camera |
| BD | Bulk density |
| c [m/s] | Speed of light |
| D | Distance from the camera to the object |
| $Dens_{burrow}$ | Burrow density |
| DSM | Digital surface model |
| $DSM_{after}$ | DSM calculated from the scan taken after the extraction |
| $DSM_{before}$ | DSM calculated from the scan taken before the extraction |
| Entrance | entrance to the animal burrow |
| g [-] | ratio [-] of the reflected photons to all photons |
| LC | National Park LC |
| LC-NL | Camera in LC on the lower north-facing hillslope |
| LC-NU | Camera in LC on the upper north-facing hillslope |
| LC-SL | Camera in LC on the lower south-facing hillslope |
| LC-SU | Camera in LC on the upper south-facing hillslope |
| MAE | Mean absolute error |
| MAP [°] | Mean annual precipitation |
| m.a.s.l. | Meters above sea level |
| MAT | Mean annual temperature |
| mClay [%] | Mean content of clay |
| $mean_{z\text{-}coordinate}$ | Mean value of the z-coordinates |



| | |
|---|---|
| **Mound** | the sediment excavated by the animal while digging the burrow |
| **mSand [%]** | Mean content of sand |
| **mSilt [%]** | Mean content of silt |
| **N** | Number of scans |
| **N** | Not affected area |
| **Not affected area** | Area not directly affected by the burrowing animal |
| **PdA** | National Park Pan de Azúcar |
| **PdA-NL** | Camera in PdA on the lower north-facing hillslope |
| **PdA-NU** | Camera in PdA on the upper north-facing hillslope |
| **PdA-SL** | Camera in PdA on the lower south-facing hillslope |
| **PdA-SU** | Camera in PdA on the upper south-facing hillslope |
| **Res** | resolution |
| **Roof** | sediment pushed aside and uphill the entrance during burrow creation |
| $S_a$ | scan after the rainfall event |
| $S_b$ | scan before the rainfall event |
| **SBC** | Single board computer |
| $sd_{z\text{-coordinate}}$ | standard deviation of the z-coordinates |
| **SSH** | Secure shell |
| **t [s]** | Overall time of camera illumination |
| **TOC [%]** | Total organic carbon |
| **ToF** | Time-of-Flight |
| $Vol_{affected}$ | volume of redistributed sediment within affected area |
| $Vol_{detected}$ | volume of the extracted sediment as detected by the camera |
| $Vol_{add}$ | difference in redistributed sediment volume between affected and not affected areas |
| $Vol_{exc}$ | Volume of the sediment excavated by the animal |
| $Vol_{hillslope\text{-}wide}$ | Hillslope-wide volume of redistributed sediment |
| $Vol_{measured}$ | volume of the extracted sediment measured by the measuring cup |
| $Vol_{per\ burrow}$ | Volume of redistributed sediment per burrow |
| $Vol_{per\ pixel}$ | Volume of redistributed sediment per pixel |
| $Vol_{redistributed}$ | volume of the calculated redistributed sediment |
| $Vol_{not\ affected}$ | volume of redistributed sediment within not affected area |
| $y_i$ | distance of the point to the point of origin at the camera nadir |
| $z_{cor}$ | Corrected z-coordinate |
| $z_{uncor}$ | Uncorrected z-coordinate |


**Table A2.** Number of usable scans for each camera

| Camera | Latitude | Longitude | Number of scans | Percentage of usable scans taken at 1am / 5am / 8am / 10pm | Time period |
|---|---|---|---|---|---|
| **PdA-NU** | -25.98131 | -70.6166 | 238 | 29 / 27 / 20 / 24 | 18.3.-18.9. |





| | | | | | |
|---|---|---|---|---|---|
| **PdA-NL** | -25.98277 | -70.61278 | 52 | 24 / 0 / 40 / 36 | 27.3.-31.5 |
| **PdA-SU** | -25.97477 | -70.61641 | 351 | 30 / 26 / 32 / 11 | 16.3.-19.9. |
| **PdA-SL** | -25.97177 | -70.61409 | 167 | 48 / 38 / 7 / 8 | 16.3.-19.9. |
| **LC-NU** | -32.95230 | -71.06231 | 215 | 37 / 20 / 8 / 33 | 9.3.-9.9. |
| **LC-NL** | -32.93928 | -71.08613 | 3 | - | 6.3.-12.9 |
| **LC-SU** | -32.93078 | -71.09066 | 160 | 22 / 28 / 26 / 25 | 28.3.-22.5 |
| **LC-SL** | -32.93110 | -71.08987 | 167 | 27 / 25 / 22 / 26 | 16.3.-19.9. |


**Table A3.** Summary of the volume of redistributed sediment, according to area and disturbance type. $Vol_{exc}$
describes volume of the sediment excavated by the animals. $Vol_{affected}$ describes volume of the sediment
redistributed during rainfall events within affected areas. $Vol_{add}$ describes the difference in redistributed
sediment volume within affected and not affected area during rainfall.

| Disturbance | Area | PdA | LC |
|---|---|---|---|
| **$Vol_{exc}$** | Affected area | 16.41 cm$^3$ cm$^{-2}$ year$^{-1}$ | 14.62 cm$^3$ cm$^{-2}$ year$^{-1}$ |
| | Per burrow | 1498.66 cm$^3$ burrow$^{-1}$ year$^{-1}$ | 1226.61 cm$^3$ burrow$^{-1}$ year$^{-1}$ |
| | Hillslope-wide | 0.18 m$^3$ ha$^{-1}$ year$^{-1}$ | 0.67 m$^3$ ha$^{-1}$ year$^{-1}$ |
| **$Vol_{affected}$** | Affected area | -1.97 cm$^3$ cm$^{-2}$ year$^{-1}$ | -10.44 cm$^3$ cm$^{-2}$ year$^{-1}$ |
| | Per burrow | -126.36 cm$^3$ burrow$^{-1}$ year$^{-1}$ | -876.38 cm$^3$ burrow$^{-1}$ year$^{-1}$ |
| | Hillslope-wide | -0.05 m$^3$ ha$^{-1}$ year$^{-1}$ | -0.48 m$^3$ ha$^{-1}$ year$^{-1}$ |
| **$Vol_{add}$** | Affected area | -1.18 cm$^3$ cm$^{-2}$ year$^{-1}$ | -7.37 cm$^3$ cm$^{-2}$ year$^{-1}$ |
| | Per burrow | -48.36 cm$^3$ burrow$^{-1}$ year$^{-1}$ | -619.2 cm$^3$ burrow$^{-1}$ year$^{-1}$ |
| | Hillslope-wide | -0.02 m$^3$ ha$^{-1}$ year$^{-1}$ | -0.34 m$^3$ ha$^{-1}$ year$^{-1}$ |



**Table A4.** Summary of the mass of redistributed sediment in Pan de Azúcar, according to area and disturbance
type. $Vol_{exc}$ describes volume of the sediment excavated by the animals. $Vol_{affected}$ describes volume of the
sediment redistributed during rainfall events within affected areas. $Vol_{add}$ describes the difference in
redistributed sediment volume within affected and not affected area during rainfall.

| Disturbance | Area | PdA | LC |
|---|---|---|---|
| **$Vol_{exc}$** | Affected area | 20.18 g cm$^{-2}$ year$^{-1}$ | 13.44 g cm$^{-2}$ year$^{-1}$ |
| | Per burrow | 1843.35 g burrow$^{-1}$ year$^{-1}$ | 1127.66 g burrow$^{-1}$ year$^{-1}$ |
| | Hillslope-wide | 246.17 kg ha$^{-1}$ year$^{-1}$ | 611.66 kg ha$^{-1}$ year$^{-1}$ |
| **$Vol_{affected}$** | Affected area | -1.73 g cm$^{-2}$ year$^{-1}$ | -9.6 g cm$^{-2}$ year$^{-1}$ |
| | Per burrow | -155.42 g burrow$^{-1}$ year$^{-1}$ | -806.26 g burrow$^{-1}$ year$^{-1}$ |
| | Hillslope-wide | -56.23 kg ha$^{-1}$ year$^{-1}$ | -436.97 kg ha$^{-1}$ year$^{-1}$ |
| **$Vol_{add}$** | Affected area | -1.45 g cm$^{-2}$ year$^{-1}$ | -6.79 g cm$^{-2}$ year$^{-1}$ |





| | Per burrow | -1.8 g burrow$^{-1}$ year$^{-1}$ | -569.65 g burrow$^{-1}$ year$^{-1}$ |
|---|---|---|---|
| | Hillslope-wide | -16.29 kg ha$^{-1}$ year$^{-1}$ | -308.57 kg ha$^{-1}$ year$^{-1}$ |


**Table A5.** Review of studies which used laser scanners for the estimation of surface processes.

| Reference | R$^2$ | Error | Horizontal point spacing | Points per cm$^{-2}$ | Model | Price |
|---|---|---|---|---|---|---|
| **Our results** | 0.77 | 0.15 cm | 0.32 cm | 8.5 | Texas Instruments OPT3101 | 900 $ |
| **(Eitel et al., 2011)** | 0.23-0.86 | 0.07 cm | NA | 25 | Leica ScanStation 2 | 102 375 $ |
| **(Eltner et al., 2013)** | NA | 0.4 cm | NA | 6.4 | Riegl LMS-Z420i | 16 795 |
| **(Kaiser et al., 2014)** | NA | NA | 0.57 cm | NA | Riegl LMS-Z420i | 16 795 |
| **(Longoni et al., 2016)** | NA | NA | NA | 1 | Riegl LMS-Z420i | 16 795 |
| **(Morris et al., 2011)** | NA | 0.5 cm | NA | NA | Maptek I-Site 4400LR | 240 000 |
| **(Nasermoaddeli and Pasche, 2008)** | NA | 0.2 cm | 0.25 cm | NA | Leica Cyrax HDS 2500 | 4500 $ |
| **(Thomsen et al., 2015)** | NA | NA | 0.4 cm | NA | Leica ScanStation 2 | 102 375 $ |


**Table A6.** Review of studies which estimated the sediment redistribution in areas affected and not affected
areas and the proposed impact.

| Reference | Climate | Animals | Method | Monitoring period | Frequency | Affected areas | Not affected areas | impact |
|---|---|---|---|---|---|---|---|---|
| **Our results** | arid | vertebrates | scanning | 7 months | Daily | 1.97 cm$^3$ cm$^{-2}$ year$^{-1}$ | 1.39 cm$^3$ cm$^{-2}$ year$^{-1}$ | +40 % |
| **Our results** | mediterranean | vertebrates | scanning | 7 months | daily | 10.44 cm$^3$ cm$^{-2}$ year$^{-1}$ | 1.39 cm$^3$ cm$^{-2}$ year$^{-1}$ | +338 % |
| **(Imeson and Kwaad, 1976)** | continental | rodents | erosion pins | 15 months | monthly | 20 mm | | NA |





| (Imeson and Kwaad, 1976) | continental | rodents | splash boards | 15 months | monthly | 91.75g 24.49 cm$^{-2}$ = 3.75 cm$^3$ cm$^{-2}$ | 94g | -3% |
|---|---|---|---|---|---|---|---|---|
| (Imeson and Kwaad, 1976) | continental | rodents | rainfall simulation (7.5 cm / hour intensity) | One-time measurement | NA | 0.2 g – 0.73 g | 0.009 g – 0.23 g | +208 % |
| (Imeson, 1977) | continental | vertebrates | rainfall simulation | One-time measurement | NA | 0.18-0.3 100 J$^{-1}$ m$^{-2}$ rain | 0.146 100 J$^{-1}$ m$^{-2}$ rain | +123 % |
| (Hazelhoff et al., 1981) | continental | earthworms | splash traps | 12 months | monthly | NA | NA | +180 % |
| (Black and Montgomery, 1991) | arid | pocket gopher | erosion pins | 10 months | 2 months | NA | NA | +125 % |
| (Hakonson, 1999) | temperate | pocket gophers | rainfall simulator (60 mm / hour) | 2 years | 2 – 3 weeks | 2.4 – 8.7 mg ha$^{-1}$ | 4.4 – 15 mg ha$^{-1}$ | -43% |
| (Li et al., 2018) | temperate | mole crickets | rainfall simulation (36 mm / hour) | One time measurement | 15 measurements | 22.1 g 115 cm$^{-2}$ = 5.2 cm$^3$ cm$^{-2}$ | 5 g 123 cm$^{-2}$ = 1.09 cm$^3$ cm$^{-2}$ | +473 % |
| (Li et al., 2018) | temperate | mole crickets | rainfall simulation (36 mm / hour) | One time measurement | 15 measurements | 35.3 g 220.5 cm$^{-2}$ = 6.24 cm$^3$ cm$^{-2}$ | 5 g 123 cm$^{-2}$ = 1.09 cm$^3$ cm$^{-2}$ | +473 % |
| (Chen et al., 2021) | lab | chinese zocor | rainfall simulation (80 mm / hour) | One-time measurement | 3 measurements | 2,69 g cm$^{-2}$ = 2.69 cm$^3$ cm$^{-2}$ | 0,88 g cm$^{-2}$ = 0.88 cm$^3$ cm$^{-2}$ | +205 % |






**Table A7.** Review of studies which estimated the sediment redistribution in areas affected by burrowing
animals, average burrow density as found in the literature and area-wide yearly contribution of burrowing
animals to sediment redistribution.

| Climate | Animals | Affected areas | Average burrow density | Average burrow size | Area-wide redistribution |
|---|---|---|---|---|---|
| **Arid** | vertebrates | 1.97 cm$^3$ cm$^{-2}$ year$^{-1}$ | 0-12 10 m$^{-2}$ = 0-1.2 m$^{-2}$ (Grigusova et al., 2021) | 91.35 cm$^2$ | 1.18 cm$^3$ ha$^{-2}$ year$^{-1}$ |
| **mediterranean** | vertebrates | 10.44 cm$^3$ cm$^{-2}$ year$^{-1}$ | 6-12 10 m$^{-2}$ = 0.6 – 1.2 m$^{-2}$ (Grigusova et al., 2021) | 84.36 cm$^2$ | 0.67 m$^3$ ha$^{-1}$ year$^{-1}$ |
| **Continental** | rodents | 91.75g 24.49 cm$^{-2}$ = 3.75 cm$^3$ cm$^{-2}$ (Imeson and Kwaad, 1976) | 14 625 m$^{-2}$ = 0.02 m$^{-2}$ (Pang and Guo, 2017) | 24.49 cm$^2$ (Imeson and Kwaad, 1976) | 0.183 m$^3$ ha$^{-1}$ year$^{-1}$ |
| **Temperate** | mole crickets | 22.1 g 115 cm$^{-2}$ = 5.2 cm$^3$ cm$^{-2}$ (Li et al., 2018) | 405 ha$^{-1}$ (Castner and Fowler, 1984) | 115 cm$^2$ (Li et al., 2018) | 0.24 m$^3$ ha$^{-1}$ year$^{-1}$ |
| **Temperate** | mole crickets | 35.3 g 220.5 cm$^{-2}$ = 6.24 cm$^3$ cm$^{-2}$ (Li et al., 2018) | 405 ha$^{-1}$ (Castner and Fowler, 1984) | 220.5 cm$^2$ (Li et al., 2018) | 0.56 m$^3$ ha$^{-1}$ year$^{-1}$ |
| **Lab** | chinese zocor | 2,69 g cm$^{-2}$ = 2.69 cm$^3$ cm$^{-2}$ (Chen et al., 2021) | 94.69 2500m$^{-2}$ = 0.04 m$^{-2}$ = 400 ha$^{-1}$ | 1256 cm$^2$ | 1.35 m$^3$ ha$^{-1}$ year$^{-1}$ |


**Table A8.** Review of studies which estimated the volume of sediment excavated by burrowing animals.

| | Climate | Animals | Method | Monitoring period | Frequency | volume of the excavated sediment |
|---|---|---|---|---|---|---|
| **Our results** | arid | vertebrates | scanning | 7 months | daily | 0.18 m$^3$ ha$^{-1}$ year$^{-1}$ |
| **Our results** | mediterranean | vertebrates | scanning | 7 months | daily | 0.67 m$^3$ ha$^{-1}$ year$^{-1}$ |



| | | | | | | |
|---|---|---|---|---|---|---|
| **(Black and Montgomery, 1991)** | arid | porcupines | mound volume | 3 years | yearly | 0.2 m$^3$ ha$^{-1}$ year$^{-1}$ |
| **(Black and Montgomery, 1991)** | arid | isopods | mound volume | 3 years | yearly | 0.11 m$^3$ ha$^{-1}$ year$^{-1}$ |
| **(Black and Montgomery, 1991)** | arid | pocket gopher | mound volume | 2 years | 3 model runs | 0.05 – 0.11 m$^3$ ha$^{-1}$ year$^{-1}$ |
| **(Rutin, 1996)** | subtropical | scorpions | mound volume | 6 months | 2-29 days | 0.42 m$^3$ ha$^{-1}$ year$^{-1}$ |
| **(Hall et al., 1999)** | alpine | rodents | mound volume | 1 year | yearly | 0.02 m$^3$ ha$^{-1}$ year$^{-1}$ |
| **(Hall et al., 1999)** | alpine | bears | mound volume | 1 year | yearly | 0.49 m$^3$ ha$^{-1}$ year$^{-1}$ |
| **(Yoo et al., 2005)** | arid | pocket gopher | mound volume | 1 year | One model run | 0.1-0.2 m$^3$ ha$^{-1}$ year$^{-1}$ |






Earth **Surface**
**Dynamics**
Discussions

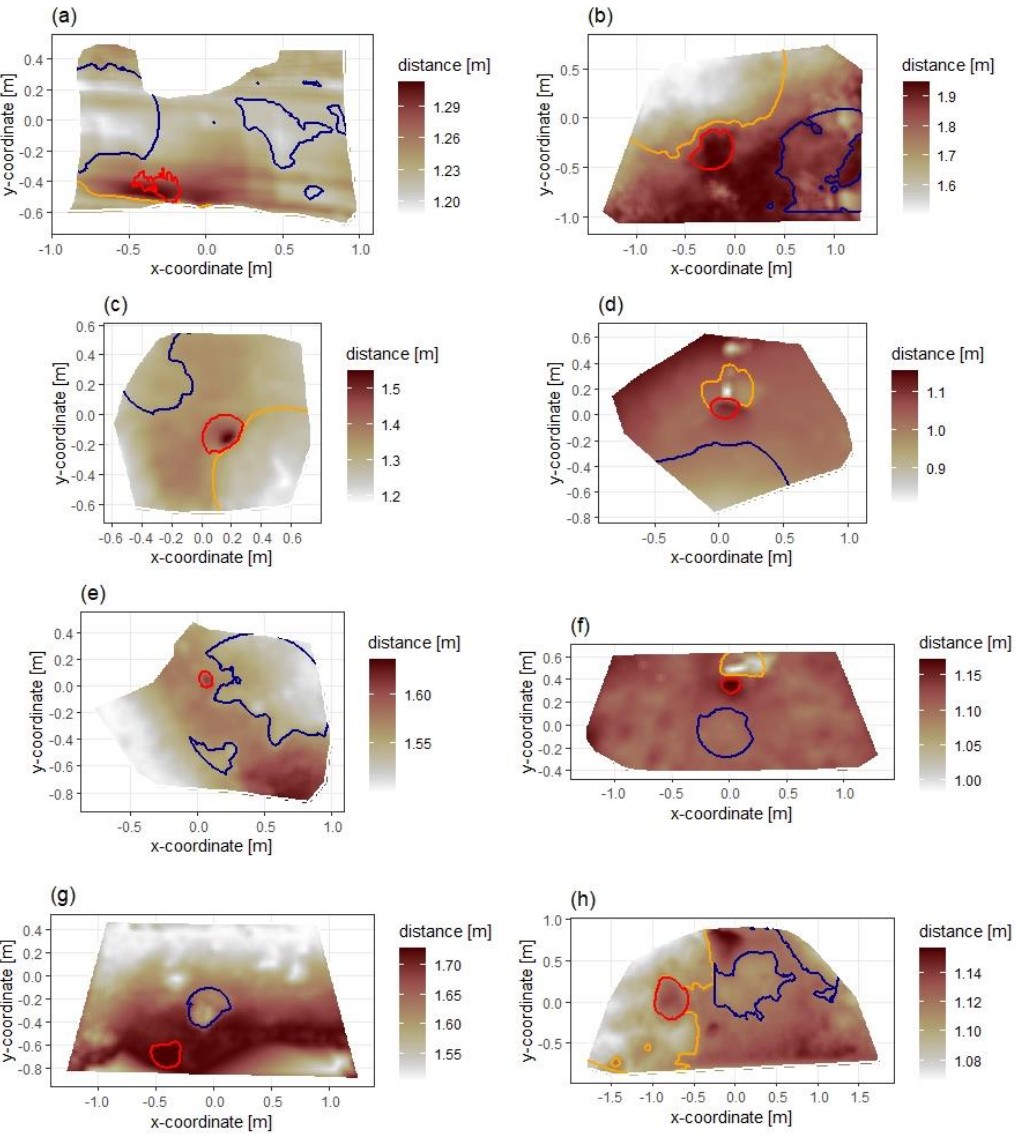

**Figure A1.** Delineation of the areas. The point of origin of the coordinate system is at the camera nadir. Depth is the distance between the surface and the camera. Red is the outline of the burrow entrance. Green is the outline of mound. Orange is the outline of burrow roof. Area which is not outlined is area not directly affected by the animal burrowing activity. Arrow indicates downhill direction of the hillslope. (a) LC-NU. (b) LC-NL (c) LC-SU. (d) LC-SL. (e) PdA-NU. (f) PdA-NL. (g) PdA-SU. (h) PdA-SL.





**Figure A2.** Sediment mass balance for the period of 7 months separately for areas affected and not affected
by burrowing animal as measured by the cameras. (a) LC-NU. (b) LC-SU. (c) LC-SL. (d) PdA-NU. (e) PdA-NL.
(f) PdA-SU. (g) PdA-SL. For abbreviations see Table A1.







738 **Figure A3.** Sediment mass balance for the period of 7 months separately for all delineated areas as measured
739 by the cameras. (a) LC-NU. (b) LC-SU. (c) LC-SL. (d) PdA-NU. (e) PdA-NL. (f) PdA-SU. (g) PdA-SL. For
740 abbreviations see Table A1.

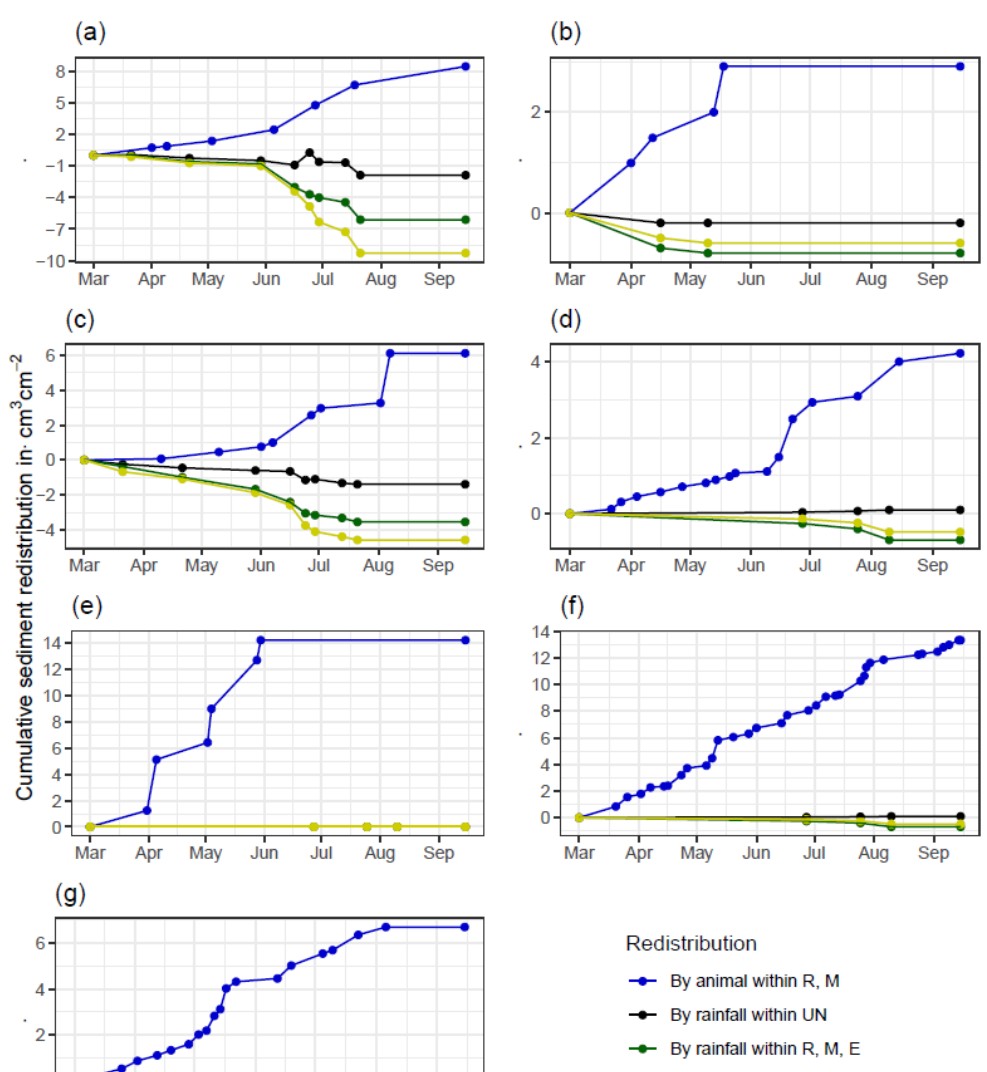

743 **Figure A4.** Cumulative volume of redistributed sediment for all cameras. Positive values indicate sediment
744 accumulation. Negative values indicate sediment erosion. Whiskers are the median sediment redistribution. E
745 is the burrow entrance. M is the mound. R is burrow roof. UN is area not directly affected by the animal
746 burrowing activity. LC is mediterranean climate zone. PdA is arid climate zone. (a) LC-NU. (b) LC-SU. (c) LC-
747 SL. (d) PdA-NU. (e) PdA-NL. (f) PdA-SU. (g) PdA-SL. For abbreviations see Table A1.

Earth **Surface** Dynamics Open Access
Discussions
EGU


**Figure A5.** Hillslope-wide volume of redistributed sediment for a time period of one year in LC. (a-d) North-
facing hillslope. (e-h) South-facing hillslope. (a) and (e) Density of burrows as estimated by Grigusova et al.
2021. (b) and (f) Volume of the sediment excavated by the animals. (c) and (g) Volume of the sediment
redistributed during rainfall events within affected areas. (d) and (h) Volume of additionally redistributed
sediment during rainfall events due to presence of the burrows. The values were calculated per burrow as
stated in section 3.7 by subtracting the sediment volume redistributed within animal affected area from the



sediment volume redistributed within not affected area and then upscaled. A stays for affected area, N stays
for not affected area by the burrowing animal.

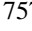


**Figure A6.** Hillslope-wide volume of redistributed sediment for a time period of one year in Pan de Azúcar. (a-
d) North-facing hillslope. (e-h) South-facing hillslope. (a) and (e) Density of burrows as estimated by Grigusova
et al. 2021. (b) and (f) Volume of the sediment excavated by the animals. (c) and (g) Volume of the sediment
redistributed during rainfall events within affected areas. (d) and (h) Volume of additionally redistributed
sediment during rainfall events due to presence of the burrows. The values were calculated per burrow as





Earth **Surface**
**Dynamics**
Discussions

stated in section 3.7 by subtracting the sediment volume redistributed within animal affected area from the
sediment volume redistributed within not affected area and then upscaled. A stays for affected area, N stays
for not affected area by the burrowing animal.

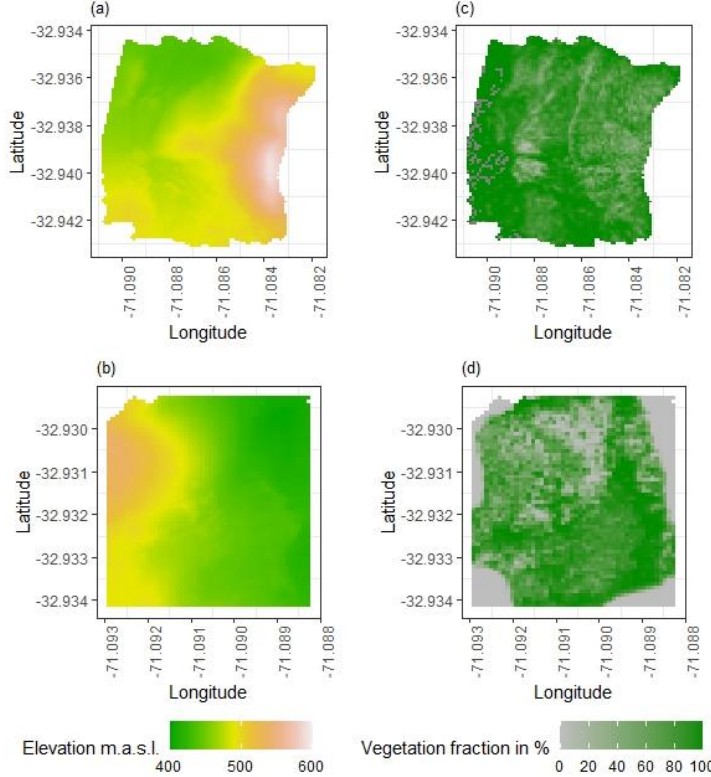


**Figure A7.** Digital surface model (a) and (b) and vegetation cover (c) and (d) of the hillslopes in LC. (a) and
(c) North-facing hillslope. (b) and (d) South-facing hillslope. m.a.s.l stands for meters above sea level.



Earth **Surface**
**Dynamics** Open Access
Discussions

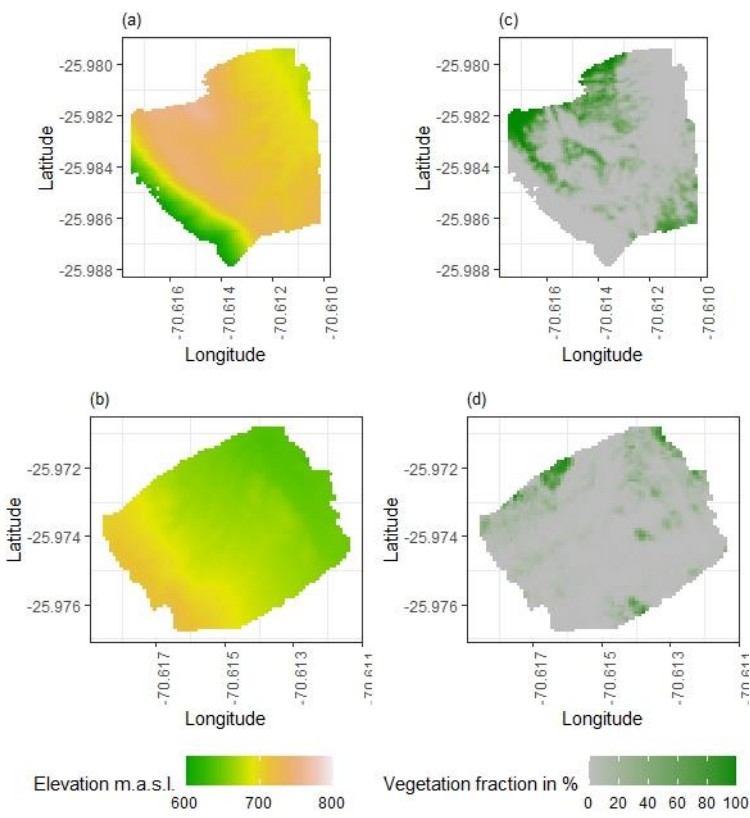


**Figure A8.** Digital surface model (a) and (b) and vegetation cover (c) and (d) of the hillslopes in Pan de Azúcar.

(a) and (c) North-facing hillslope. (b) and (d) South-facing hillslope. m.a.s.l stands for meters above sea level.



















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
