# Peer review of "Higher sediment redistribution rates related to burrowing animals than previously assumed as revealed"

_Earth Surface Dynamics, 2021_

## Author Comment (AC1)

**Response to reviewer 1:**

**[R1C1]:**

I like this study and I like the presentation of this novel method to measure soil redistribution. But…I am not sure I agree with all aspects of the application of this technique and the emphasis in terms of objectives in this study.

**[R1R1]:**

Thank you for your thoughtful review that has improved the work. We believe we have addressed all comments and modified the manuscript accordingly

**[R1C2]:**

I suggest splitting the paper, with one paper being a short methodological paper, with all the technical details of the method, and accuracy testing (I like how that was approached). And the other focusing on this comparison between arid and Mediterranean, using a clearly phrased research question, such as the one that I suggest. But, I would be careful in how to approach answering this question. In general, I thought the article was quite long, with many relatively complicated diagrams to follow. Can this be simplified? I think splitting the paper in two will already help with this.

**[R1R2]:**

Our study combines new methods and demonstration of applicability to highlight both the potential and relevance of the approach to monitoring biogeomorphological processes. We understand where the suggestion of the reviewer is coming from, however we wish to preserve the paper without splitting to ensure the method and application are retained, as this appeals to a larger range of researchers. This approach is also consistent with broader editorial norms e.g. Lane, 2017 (DOI: 10.1111/cag.12329).

However, we understand the reviewer's concern regarding paper length and density. We have tried to address this by significantly simplifying and shortening the manuscript by removing 3 figures and shortening the manuscript by 4 pages

Specifically, we have removed figures 4 and 7. Additionally, we shortened the result section which now consists of only 3 subsections (previously was 5 sections). We also removed 2 paragraphs and joined two subsections together. Now, the first section only very briefly describes the camera accuracy as well as the amount and the quality of scans for each camera. The second subsection describes the daily mass balance and cumulative volume of redistributed sediment. The third subsection consists of the overall volume of redistributed sediment.

**[R1C3]:**

How unaffected are the "unaffected" sites at this small scale then really? Surely the burrowing animals walk around in between their burrows, trampling the soil. Essentially, is the sediment redistribution in the non-affected areas representative of sites that do not have burrowing animals at all? If we took away the animals, would the situation in the non-affected sites be representative of how much sediment was redistributed? This has to be discussed a bit or acknowledged if there is doubt.

**[R1R3]:**

We agree with the reviewer and have modified the manuscript accordingly.

Lines 311-313: "Please note that the areas termed "not affected" by the burrowing animals are areas adjacent to burrows. This does not imply complete absence of animals, just no active burrowing.

In the for this study relevant time period (daily-yearly), the manuscript aims to quantify animal impact that is measurable on a mm scale. On this scale, areas that show animal-related surface change are classified as 'affected', and all other areas which are not directly impacted by animals are classified as

'unaffected'. These are defined in lines 273-281 as follows: "The affected areas included three sub-areas: (i) mound (M), (ii) entrance (E) and (iii) burrow roof (R). "Mound" describes the sediment excavated by the animal while digging the burrow. "Entrance" describes the entry to the animal burrow up to the depth possible to obtain via the camera. "Burrow roof" describes the part of the sediment above and uphill the burrow entrance (Bancroft et al. 2004). The remaining surface within the camera's FOV, which shows no surface change due to animal impact, was classified as not affected (N) by the burrowing animal during the creation of its burrow. After this short description, we use 'unaffected' and 'affected' throughout the manuscript instead of '(un)affected by burrowing animals' to ensure the readability of the manuscript. We have now clarified the reasoning behind the set-up of our study:

**[R1C4]:**

In terms of the temporal upscaling (from seven months to a year), I am also not convinced. Burrowing animals do not continue burrowing at the same pace. And presumably the authors did not catch the burrowers right in the beginning, when the burrowing first started and was at its fastest. I was also missing information on who these burrowers are. What sort of animals are we talking about? This would influence how realistic upscaling is. Again, I also don't think they need to upscale temporally to compare the two systems.

**[R1R4]:**

Thank you for this comment. All burrowing animals present at our sites are listed in the section below.

Lines 150-153: "Among the most common vertebrate burrowing animals in PdA are carnivores *(Lycalopex culpaeus, Lycalopex griseus),* marsupials and rodents *(Phyllotis xanthopygus, Phyllotis limatus, Abrothrix andinus)* (Jimenez et al. 1992; Cerqueira 1985). In LC these are rodents *(Octodon degus, Rattus norvegicus and Phyllotis darwini)* and carnivores *(Lycalopex griseus)* (Muñoz-Pedreros et al. 2018)."

Secondly, with regards to the temporal upscaling, we now included the calculated sediment redistribution for the period of 7 months into the manuscript (Table A4, lines 667-671, see below). However, to enable comparability of our study with all other studies in this field of research, we also give the volume of redistributed sediment for the period of one year throughout the manuscript. Please note that this calculation is rather a change of the units of a calculated rate (mm/7 months --> mm/yr) than upscaling, which assumes that the rates are valid over the period without measurement. This is a very common practice that is critical for placing the work in a wider context, but we emphasise the caveat inherent in the assumptions of the timescale change over which the rates are estimated. We are confident in the timescale extrapolation of the rate because:

i) In contrast to previous studies, our study provides daily data on sediment redistribution which allow a more realistic temporal upscaling than the data sampling with lower frequency.

ii) All previous studies estimated the volume of redistributed sediment per year, even though the measurements were conducted once a year, and thus completely ignored the ongoing sediment excavation and erosion processes

iii) Our study was conducted from middle autumn to middle of spring and thus covered exactly half of the vertebrate burrowing season (Romanach et al. 2005), including dry and wet seasons, thus capturing the key cycles of variability.

iv) One of the main goals of our study is to calculate more precisely the impact of burrowing animals on sediment redistribution than it was done in all previous studies. This is only possible when we compare our results with the results of previous studies. These stated the sediment volume, as already, all presented in $m^3$ year$^{-1}$ regardless the frequency and duration of measurements.

v) We understand that especially in the arid climate, one would ideally like at least 3 years of data due to the infrequent rainfall. We are not aware of any study which has delivered this type of data. Thus, we believe our far higher frequency measurements provide detailed insights on redistribution processes within the dry periods, which has also never been done before. We included this in the manuscript (Lines 259-260): "The cameras collected the data for the time period of 7 months.

vi) We also included a short paragraph to make it absolutely clear to the reader that the yearly rates are upscaled values:

Lines 397-398:

"Please note that we used the volume of redistributed sediment monitored for 7 months to calculate the volume of sediment per year."

Lines 667-670:

**Table A4.** Summary of the volume of redistributed sediment for the period of 7 months, according to area and disturbance type. $Vol_{exc}$ describes volume of the sediment excavated by the animals. $Vol_{affected}$ describes volume of the sediment redistributed during rainfall events within affected areas. $Vol_{add}$ describes the difference in redistributed sediment volume within affected and not affected area during rainfall.

| Disturbance | Area | PdA | LC |
|---|---|---|---|
| $Vol_{exc}$ | Affected area | 9.57 $cm^3$ $cm^{-2}$ 7 months$^{-1}$ | 8.53 $cm^3$ $cm^{-2}$ 7 months$^{-1}$ |
| | Per burrow | 874.22 $cm^3$ burrow$^{-1}$ 7 months$^{-1}$ | 715.52 $cm^3$ burrow$^{-1}$ 7 months$^{-1}$ |
| | Hillslope-wide | 0.11 $m^3$ ha$^{-1}$ 7 months$^{-1}$ | 0.39 $m^3$ ha$^{-1}$ 7 months$^{-1}$ |
| $Vol_{affected}$ | Affected area | -1.15 $cm^3$ $cm^{-2}$ 7 months$^{-1}$ | -6.09 $cm^3$ $cm^{-2}$ 7 months$^{-1}$ |
| | Per burrow | -73.71 $cm^3$ burrow$^{-1}$ 7 months$^{-1}$ | -511.22 $cm^3$ burrow$^{-1}$ 7 months$^{-1}$ |
| | Hillslope-wide | -0.03 $m^3$ ha$^{-1}$ 7 months$^{-1}$ | -0.28 $m^3$ ha$^{-1}$ 7 months$^{-1}$ |
| $Vol_{add}$ | Affected area | -0.69 $cm^3$ cm$^{-2}$ 7 months$^{-1}$ | -4.30 $cm^3$ $cm^{-2}$ 7 months$^{-1}$ |
| | Per burrow | -28.21 $cm^3$ burrow$^{-1}$ 7 months$^{-1}$ | -361.20 $cm^3$ burrow$^{-1}$ 7 months$^{-1}$ |
| | Hillslope-wide | -0.01 $m^3$ ha$^{-1}$ 7 months$^{-1}$ | -0.2 $m^3$ ha$^{-1}$ 7 months$^{-1}$ |

**[R1C5]:**

How large a "scan" was, i.e. what are they upscaling from? Or how uniform the burrow sizes at any given point in time were? Burrows in the landscape are presumably at different stages of creation – most natural systems are dynamic systems with burrows being created and destroyed by rainfall continuously. What about other non-burrowed features in the landscape? Big rocks or trees etc. Are the scans and the non-burrowed areas within them really representative of the landscape? Can one just upscale from one burrow to a whole slope of burrows? I also don't think it's necessary to upscale to answer the interesting questions in this study. That can be done at the smaller scale of individual burrows and non-burrowed areas

**[R1R5]:**

Regarding the scan size, this is mentioned in line 285 and is 4 $m^2$. The spatial upscaling is based on an earlier, already published manuscipt by Grigusova et al., 2021 (https://doi.org/10.3390/drones5030086), which focuses solely and very detailed on modelling the burrow density using random forest in the same research areas. Based on the earlier estimated amount of burrows per pixel, we calculate the hillside-wide volume of animal-caused and rainfall-caused sediment redistribution per 100 $m^2$. We have tried to clarify and shorten the passages on this topic, but retainedthe calculation as this gives the possibility unravel some tendencies in animal-related sediment transport that would otherwise have been overlooked. We e.g. found that while on the burrow scale the animal-caused redistribution is higher in PdA, on the hillside-scale (and thus when one wants to look on the overall role of burrowing animals on the ecosystem processes) it is higher in LC, which is due to higher burrow density in LC. This point would not be clear to the reader if we would delete this section, and provide only data on the burrow scale. Please see lines 496-498:

Lines 493-495: "The volume of the sediment redistributed by the animal was lower in LC than in PdA (Fig. 9, Table 1). However, on the hillslope scale, a higher total area-wide volume of excavations was calculated for LC compared to PdA, due to the higher burrow density in LC."

**[R1C6]:**

Other interesting research questions that can potentially be answered with the data set are listed below.

1. How variable are the burrow sizes in this landscape?
2. Does this differ between the arid and Mediterranean system?
3. How fast do the animals burrow?
4. How variable is this?
5. How does this change over time?
6. How fast do the burrows deteriorate after a rainfall event?

**[R1R6]:**

These are indeed interesting questions, which we answered in the manuscript as follows:

Lines 56-57: The animals burrowed between on average 1.2 – 2.3 times a month and the burrowing intensity increased after rainfall.

Lines 129-130: We estimated the burrowing intensity and its dependence on rainfall.

In the result section, we then provided answers to these questions:

1. How variable are the burrow sizes in this landscape?
2. Does this differ between the arid and Mediterranean system?

Lines 485-486: "The average size of the burrows was 84.36 $cm^2$ (SD = 32.54 $cm^2$) in LC and 91.35 $cm^2$ in PdA (SD = 8.53 $cm^2$).

3. How fast do the animals burrow?

Lines 486-488: The animals burrowed (increased the size of the particular burrow) on average 1.2 times $month^{-1}$ in LC and 2.33 times $month^{-1}$ in PdA. The volume of the excavated sediment was 102.22 $cm^{-3}$ $month^{-1}$ in LC and 124.89 $cm^3$ $month^{-1}$ in PdA. Each time the animals burrowed, they excavated on average 42 $cm^3$ sediment volume in LC and 14.33 $cm^3$ sediment volume in PdA.

4. How does this change over time?

Lines 489-490: The burrowing intensity increased in winter after the rainfall occurrences in LC and stayed constant during the whole monitoring period in PdA.

5. How fast do the burrows deteriorate after a rainfall event?

Lines 490-491: The burrows deteriorate after rainfall events with a rate of 73.03 $cm^3$ $month^{-1}$ or 63.90 $cm^3$ $event^{-1}$ in LC and 10.53 $cm^{-3}$ month or 24.57 $cm^3$ $event^{-1}$."

**[R1C7]:**

What is meant with autonomous in this context? Do they mean automated?

**[R1R7]:**

Yes, we meant the term "automated". We corrected the term in all cases (Lines 43, 116, 125 and 500).

**[R1C8]:**

Lines 49 to 50: This is exactly where it would be interesting to tease apart how much of this was a result of burrowing and how much was rainfall? Both the rainfall and the burrowing species presumably differ between the systems. This is an interesting question to phrase the whole project around. At the moment, the fact that they had two different systems is almost an "aside".

**[R1R8]:**

Thank you very much for this suggestion, we followed your recommendation and reconstructed the paragraph in the abstract as follows:

Original paragraph:

"The cumulative sediment redistribution within areas affected by burrowing animals was higher (-10.44 $cm^3$ $cm^{-2}$ $year^{-1}$) in the mediterranean than the arid climate zone ( -1.41 $cm^3$ $cm^{-2}$ $year^{-1}$)."

Rephrased:

Lines 49-52:

The animal-caused cumulative sediment redistribution was 14.62 $cm^3$ $cm^{-2}$ $year^{-1}$ in the mediterranean and 9.57 16.41 $cm^3$ $cm^{-2}$ $year^{-1}$ in the arid climate zone. The rainfall-caused cumulative sediment redistribution within areas affected by burrowing animals was higher (-10.44 $cm^3$ $cm^{-2}$ $year^{-1}$) in the mediterranean than the arid climate zone ( -1.41 $cm^3$ $cm^{-2}$ $year^{-1}$).

**[R1C9]:**

Consider adding a study species section after the study site section.

**[R1R9]:**

We agree with the reviewer and added a paragraph about the composition of present species (Lines 150-153)

Lines 150-153: "Among the most common vertebrate burrowing animals in PdA are carnivores *(Lycalopex culpaeus, Lycalopex griseus),* marsupials and rodents *(Phyllotis xanthopygus, Phyllotis limatus, Abrothrix andinus)* (Jimenez et al. 1992; Cerqueira 1985). In LC these are rodents *(Octodon degus, Rattus norvegicus and Phyllotis darwini)* and carnivores *(Lycalopex griseus)* (Muñoz-Pedreros et al. 2018)."

**[R1C10]:** Is Figure 4 necessary?

**[R1R10]:** We had to compromise here between your comment and the comment of the reviewer #2 - and decided to moved figure 4 to Appendix, which is now Figure A3 (Line 698).

**[R1C11]:** Line 403: Exemplarily is not a word.

**[R1R11]:** Thank you for picking up on this – we changed the word to exemplary.

Response to Reviewer 2:

**[R2C1]:**

The study illustrates an interesting application of time-of-flight cameras in geomorphology. Furthermore, it very well highlights the potential of custom build sensor systems with simple components (i.e. Pi) with full automatic/autonomous capabilities. The manuscript is well structured and easy to follow. I agree with the first reviewer that the already captured data entails enough novel information to present in ESurf. However, some issues regarding the methods should be addressed in more detail, which are displayed in detail below:

**[R2R1]:**

Dear reviewer #2,

That you very much for your helpful review. We addressed all of your comments and included the required information in the manuscript. Please find our point to point answer to your review below. You find reviewer comments underlined, our answers in black, the pre-reviewed manuscript text in blue, and the modified manuscript text in green.

**[R2C2]:**

Chapter 1: What is actually a low-cost ToF? No prices (or at least rough estimates) are mentioned and therefore the statement low-cost is not possible to assess. The authors discuss the drawback of laserscanning, as being a lot more expensive. However, laserscanning also reaches a lot farther compared to the ToF cameras. Therefore, the types of studies that can be performed are not relatable due to the different observation scales. The authors miss mentioning time-lapse photogrammetry as another already applied low-cost (as even track-cameras might be used) topographic monitoring technique that can be applied at different observation distances and thus scales (e.g. James et al., 2014 and Galland et al., 2016 – volcanology, Eltner et al., 2017 – soil erosion, Mallalieu et al., 2017 – glacier, Kromer et al., 2019 and Blanch et al., 2021 – rock falls).

**[R2R2]:**

We agree with the reviewer, and added a paragraph on comparing TOF with time-lapse photogrammetry, plus incorporated studies on time lapse photogrammetry in the introduction as well as in the discussion. We also deleted parts of the discussion regarding laser scanning.

Lines 116-123: "An already applied low-cost (up to 5000$) topographic monitoring technique is time-lapse photogrammetry which can be applied at variable observation distances and scales (e.g. (James und Robson 2014; Galland et al. 2016; Eltner et al. 2017; Mallalieu et al. 2017; Kromer et al. 2019; Blanch et al. 2021). For this technique, the surface has to be monitored under various angles for which several devises are needed to be installed in the field. In contrast, Time-of-Flight (ToF) technology offers a new cost-effective possibility for a high-resolution monitoring of sediment redistribution (Eitel et al. 2011; Hänsel et al. 2016) which can be achieved by a simple installation of one devise in the field."

Lines 536-540: "The estimated costs in studies using time-lapse photogrammetry were similar to our study (James and Robson, 2014; Galland et al., 2016; Mallalieu et al., 2017; Eltner et al., 2017; Kromer et al., 2019; Blanch et al., 2021)."

**[R2C3]:**

Line 162-165: I find the explanation of the pulsed ToF principle confusing. It should be added that the receiver is opening the first window simultaneously and synchronised with the pulse emission, i.e. the receiver opening the window with the same delta t as the emitted pulse. And then the second window is opened, for the same duration delta t, synchronised with the closing of the first window. Thus, the captured photon number (i.e. measured by electrical charge) in both windows can be related according to equation 1 (the higher g1 the shorter the distance) to solve for the distance, which can also be considered as solving for the phase shift and thus solving the ToF. Maybe, the authors can also shortly mention that in general the ToF cameras rely on the principle of measuring the phase shift and that there are different

options to modulate the light source to be able to measure a phase shift, e.g. the camera in this study using pulsed modulation.

**[R2R3]:**

We agree with the reviewer and explain the Time-Of-Flight principle clearer plus added the required paragraph into the manuscript as follows:

Lines 165-175: ""ToF cameras rely on the principle of measuring the phase shift, with different options to modulate the light source to measure it. The cameras employed used pulse-based modulation, meaning the light pulse was first emitted by the camera, then reflected from the surface and finally measured by the camera using two temporary windows. The opening of the first window is synchronized with the pulse emission i.e. the receiver opens the window with the same $\Delta t$ as the emitted pulse. Then, the second window is opened, for the same duration $\Delta t$, which is synchronised with the closing of the first window. The first temporary window thus measures the incoming reflected light while the light pulse is also still emitting from the camera. The second temporary window measures the incoming reflected light when no pulse is emitting from the camera. The captured photon number (i.e. measured by electrical charge) in both windows can be related according to equation 1, and the distance from the camera to the object can then be calculated as follows:"

**[R2C4]:**

Line 172-173: The spatial resolution also depends on the orientation of the camera. The more oblique the perspective, the more the variation.

**[R2R4]:**

Thank you, we added the statement to the manuscript:

Line 182-183: "The camera's field of view (FOV) and the spatial resolution of the scans depended on the height of the camera above the surface and camera orientation."

**[R2C5]:**

Line 174-175: The point cloud can be both binary and encoded. The authors are actually describing a cloud stored in a binary format being transformed into an ASCII (?) encoded data format.

**[R2R5]:**

Thank you, yes, that was our intention. We corrected the sentence:

Lines 184-185: "The point clouds taken by the camera were transformed from the binary format to an ASCII format."

**[R2C6]:**

Line 176-179: As I understand this, the centre of the camera sensor defines the origin of the local, 3D Cartesian coordinate system?

**[R2R6]:**

Thank you, yes, it does. We explained this clearer in the manuscript:

Lines 186-188: "The coordinates were distributed within a three-dimensional Euclidian space, with the point at the camera nadir (the centre of the camera sensor) being the point of origin of the 3D Cartesian coordinate system."

**[R2C7]:**

Equation 2: What is actually wrong with the original Z-value?

**[R2R7]:**

Thank you for your question. The z-coordinate describes the distance from the surface to the camera. As the camera was titled 10° and the burrow was located on a hill, meaning the surface was also titled, by not correcting the z-coordinate, the volume of the burrow located downhill the camera would seem to be under- or overestimated. For an easier calculation of surface changes, we projected the burrow on a flat surface.

**[R2C8]:**

Are the authors aiming to transform the measurements to a local coordinate system, where the X-Y-axes are parallel to the soil surface (for the subsequent transformation of Z-values to a 2.5D dataset)? If yes, would a simple rigid body transformation not be enough?

**[R2R8]:**

Thank you for raising this this point, but no, we did not aim to transform the Z-values to a 2.5D dataset.

**[R2C9]:**

Furthermore, why is the distance of a distance, i.e. distance($y_1$-$y_i$), calculated? Do you mean solely ($y_1$-$y_i$)?

**[R2R9]:**

We meant solely ($y_1$-$y_i$). We corrected the equation accordingly (Line 195):

$$z_{cor} = z_{uncor} - \tan(\alpha + \beta) * (y_1 - y_i) \qquad . \qquad\qquad\qquad (2)$$

**[R2C10]:**

Also, if the authors refer to the distance of the origin, thus the radius, I would suggest to use $r_{xy}$ (sqrt(sqr($x_1$-$x_i$)+sqr($y_1$-$y_i$))) instead of y. This causes confusion, as y is already explained as the y-coordinate.

**[R2R10]:**

We did not mean the distance to the point of origin, but the difference in y-coordinates between each point, and the point with an y-coordinate = 0 and the same x-coordinate as the respective point. We changed the manuscript accordingly:

Lines 196-199: "In Eq. (2), $z_{cor}$ is the corrected distance (m) between the camera and surface (m), $z_{uncor}$ is the uncorrected z-coordinate (m), α is the tilt angle of the camera (°), β is the surface inclination (°), and $y_i$ (m) is the distance between each point, and the point with i) an y-coordinate = 0 and ii) the same x-coordinate as the respective point."

**[R2C11]:**

How did the authors calculate the angles and with what accuracies? This seems to be tricky in the field.

**[R2R11]:**

Thank you for pointing this out. To estimate the inclination, we used the digital Clinometer from plaincode which has an accuracy of 0.1 degrees. We measured the surface inclination next to the uppermost and lowermost part of the burrow and calculated the average inclination.

**[R2C12]:**

Equation 3: Why did the authors choose the scaling of 1 standard deviation and not e.g. 1,5 or 2?

**[R2R12]:**

We tried different methods and threshold values to correct the scattering and remove the erroneous data points. We visually validated the corrections each time. The proposed method provided the best results, meaning all erroneous data points were removed while the correct data points were kept. When using a different threshold, either several erroneous data points were not detected as erroneous by the algorithm or several correct data points were removed, up to the point that several otherwise correct scans would not contain enough data points for analysis.

**[R2C13]:**

Chapter 3.3: Why did the authors not compare the ToF data in the lab experiment with SfM (sub-mm accuracy at that close range possible) or a triangulation based LiDAR (μm accuracy possible)? Such references allow the assessment of spatially distributed errors or potential spatially correlated errors. If the authors use SfM they could have also done an accuracy assessment outside under the actual observation conditions.

**[R2R13]:**

We understand the importance to assess the spatial distribution of errors. However, to do this, a comparison with other techniques didn't seem to be necessary in our case, as the error clearly increased from the centre of the scan towards the corners or with the distance from the camera nadir. This was clearly visible also looking at the raw data / scans before any processing as the z-coordinate values of the points at scan corner deviated from the z-coordinate values in the centre. It can also be seen in Figure A1 showing the spatially distributed standard deviation of the z-coordinates of two scans showing the same surface – the standard deviation clearly increases toward the scan corners. The standard deviation was here calculated from scans before any corrections. During the processing, we cropped the scans and removed highly scattered points as according to the chapter 3.2.

We included a description of the error spatial distribution in the manuscript as follows:

Lines 223-224: "The deviation increases from the centre /camera nadir towards the corners of the scan."

**[R2C14]:**

Line 209-211: I understand that the authors average the data from several subsequent scans to reduce the noise, assuming a random (Gaussian distributed) error. However, in regard of the accuracy estimation, I would suggest to display the standard deviation, also spatially distributed, to get a better grasp on the variation of each scan.

**[R2R14]:**

Thank you for your suggestion. We calculated the standard deviation of the two scans and included a figure showing the spatially distributed standard deviation. The changes were made in the manuscript as follows:

Lines 222-223: "The standard deviation of the z-coordinate of the two scans taken each time was 0.06 cm. Figure A1 shows the spatially distributed standard deviation."

[Figure]

Line 686-688: "Figure A1. Standard deviation (SD) of the z-coordinate of unprocessed five scans showed exemplary for the camera on the upper north-facing hillside. The average standard deviation was 0.06 cm. The reader may note an increase of the standard deviation toward the corners of the scan."

**[R2C15]:**

Line 211-212: How did the author assure a smooth surface? What was the surface made of? I suggest, instead of using the standard-deviation of the Z-coordinate as error estimate (which will be overestimated if the surface is tilted), to fit a plane into the point cloud and calculate the distance to that surface to get the variation in the distance measurements.

**[R2R15]:**

Thank you for raising this point. The surface was the linoleum floor within our office and the scan was taken in the night with the lights off. We understand your point regarding the Z-coordinate. To ensure that the standard deviation of the Z-coordinate would not be overestimated, we correct the Z-coordinate in Equation 2. Nevertheless, we followed your instruction, fitted the plane into the point cloud and calculated the distance. The variation in the distance measurements was 0.17 cm. We took this as our new threshold value and repeated the test, however, the accuracy didn't improve. We included the new way of calculating the threshold value in the manuscript as follows:

Lines 226-230: "A scan was taken of a smooth surface (linoleum floor) and a point cloud was created from the data. Then, we fitted a plane into the point cloud and calculated the distance between the plane and the camera sensor. The standard variation (0.17 cm) in the distance measurements was saved. Solely, the differences between the DSMs below this variation were considered in the calculation of the detected sediment extraction."

**[R2C16]:**

Line 223: Please, also display the standard deviation to assess the random error and potentially display a boxplot to better illustrate the inherent variability in your method as you have 45 measurements allowing for such a display.

**[R2R16]:**

We calculated the standard deviation and created a boxplot as required. We changed the manuscript as follows:

Lines 402-403: "The accuracy between the measured extracted sediment volume and sediment volume calculated from the camera scans was very high (MAE = 0.023 cm$^3$ cm$^{-2}$, R$^2$ = 0.77, SD = 0.02 cm$^3$ cm$^{-2}$, Fig. A3)."

As the reviewer #1 suggested to remove Figure 4, we compromised between the two comments and moved the figure to the appendix as figure A3 (Line 698-704):

[Figure]

"Figure A3. a) Estimation of Time-of-Flight camera accuracy based on averaging two surface scans before and after the sediment extraction under controlled conditions. The x-axis shows the exact sediment volume measured with a cup. The y-axis represents the volume of the sediment calculated from the camera scans (according to Equation (4)). The blue line is the linear regression calculated from the measured and detected volume. The green shadow shows the confidence interval of 95% for the linear regression slope. ***p ≤ 0.001. MAE is the mean absolute error, SD is standard deviation and R$^2$ the coefficient of determination. b) Measured sediment volume subtracted from the detected sediment volume for all measurements."

**[R2C17]:**

Chapter 3.5: The choice of the parameters to derive the entrance or mound seem arbitrary. The motivation and reasoning for the choices as well as the defined thresholds should be explained in more detail.

**[R2R17]:**

Thank you for bringing up this discussion. We added the motivation for the defined thresholds to the manuscript:

Lines 301-310: "We used the DEM and slope layers for the delineation for several reasons. The distance from the surface to the camera was the most important parameter to derive (i) the deepest point of the entrance and (ii) the highest point of the mound or burrow roof, as this was (mostly) the closest point to the camera. After the angle correction of the z-coordinate according to chapter 3.2., the surface inclination of the areas without burrow was 0°, while the angle between the border of the burrow entrance or mound and the not-affected surface was above 0°. Because neither the entrance nor the mound have a perfect circular form, we would largely overestimate or underestimate the entrance or mound size. Overestimate by not stopping the search algorithm until the angle between all new points of the buffer to the rest of the buffer was 0°. Underestimate by stopping the algorithm when the angle of one point of the buffer to the nearest point of the buffer was 0°. The value of 50% thus minimized the error."

**[R2C18]:**

Line 266-267: What is the spatial resolution of the DSM?

**[R2R18]:**

We included this information in the manuscript (Line 283):

"The DSM had a spatial resolution of 0.6 cm."

**[R2C19]:**

Line 274: Why 16 squares and what was their size?

**[R2R19]:**

Originally, we intended to divide the scan into 4 squares for the four quadrants within the 2D grid (x- and y-axis). As the algorithm did not detect the burrow position correctly this way, we increased the number of squares and divided each of the quadrants into 4 squares. The squares had a size of 0.5 m². We did not need to increase the number of squares to 32 as with 16 squares all of areas were identified correctly. We explained this better in the manuscript:

Lines 290-292: "Both the uphill and the downhill parts were subdivided into 16 squares, so that each of the four quadrants within the 2D grid (x- and y-axis) contained four squares. The squares had a size of 0.5 m²."

**[R2C20]:**

Line 297-298: What is the standard deviation of the five scans? This could also be used to assess the accuracy of the measurements?

**[R2R20]:**

Lines 358-359: "The average standard deviation of the z-coordinate of these scans was 0.06 cm."

**[R2C21]:**

Line 323: Why five scans? Did the authors test that at this number, accuracy does not increase much more after averaging? Or is this due to storage or power consumption

**[R2R21]:**

We first tested the accuracy using just one scan before and one scan after then sediment extraction. Then, we increased the number of scans and averaged them. However, the accuracy did not increase much after averaging more than two scans. However, we decided to save five scans per measurement in the field to ensure we will have at least two scans for each time slot in case some of the scans were not usable. We decided to take five scans per time slot and not more due to storage capacity of the field device.

**[R2C22]:**

Chapter 3.7: How did the author ensure that there is no mixture/overlap of different processes, e.g. erosion due to rainfall happening shortly after digging?

**[R2R22]:**

Our cameras captured the data four times a day (approximately every six hours). The precipitation data were averaged per hour. We derived the rainfall-caused and animal-caused sediment redistribution by comparing two following scans. If a rainfall event occurred within these six hours, the redistribution within all areas was thought to be due to rainfall event. If during these six hours digging occurred, then the mound and roof height increased, depth of the entrance tunnel increased and there were no changes

within not affected areas. If both processes took place during six hours, the following conditions would have applied: i) rainfall event occurred, ii) burrow size changed as after digging, iii) sediment eroded from not affected areas. We could not differentiate, if the rainfall-caused or animal-caused redistribution occurred first during these hours. This case however, never occurred. We added this information into the manuscript:

Lines 346-348: "If both animal-caused and rainfall-caused sediment redistribution took place, the following conditions applied: i) rainfall event occurred, ii) burrow size changed as after digging (mound height increased, entrance depth increased, burrow roof height increased or decreased), iii) sediment eroded from not affected areas."

**[R2C23]:**

Line 337: What machine learning algorithm was used? After checking the cited paper, I understood that an in the other study trained random forest was used again in this study. I would suggest to add this information; thus others can follow the manuscript without needing to check the references.

**[R2R23]:**

We agree with the reviewer and changed the manuscript accordingly:

Lines 387: "For the upscaling, we applied a random forest model with recursive feature elimination."

**[R2C24]:**

Equation 7: What is M? Did you mean Vol?

**[R2R24]:**

Yes, we did. We changed the abbreviation accordingly (Line 375):

$$\mathrm{Vol_{add}} = (\mathrm{Vol_{affected}} - \mathrm{Vol_{unaffected}}) \qquad\qquad , \qquad\qquad\qquad (7)$$

**[R2C25]:**

Equation 7-10: The authors observed the sites solely for 7 months and upscale then to yearly changes. Can this be done so easily. For instance, at the mediterranean site at least a full year should be observed to capture all the seasons. For the desert site the observation period would need to be even longer.

We understand your concerns with the temporal upscaling from 7 months to one year. We now included the calculated sediment redistribution for the period of 7 months into the manuscript.

**[R2R25]:**

However, to enable comparability of our study with previous studies, we also still show the volume of redistributed sediment for the period of one year. We included a paragraph pointing out a possible uncertainty of the upscaled values in the methodology section:

Lines 397-398:

"Please note that we used the volume of redistributed sediment monitored for 7 months to calculate the volume of sediment per year."

We decided to keep the temporal upscaling for several reasons:

    i)        In contrast to previous studies, our study provides daily data on sediment redistribution which allow a more realistic temporal upscaling than the data sampling with lower frequency. Previous studies published in EGU journal Biogeography by Übernickel et al.

2021 measured the volume of excavated sediment solely once and stated that this is a yearly sediment excavation by the animals

ii) All previous studies estimated the volume of redistributed sediment per year, even though the measurements were conducted once a year, and thus completely ignored the ongoing sediment excavation and erosion processes

iii) Our study was conducted from middle autumn to middle of spring and thus covered exactly half of the vertebrate burrowing season (Romanach et al. 2005)

iv) One of the main goals of our study is to calculate more precisely the impact of burrowing animals on sediment redistribution than it was done in all previous studies. This is only possible when we compare our results with the results of previous studies. These stated the sediment volume, as already, all presented in $m^3$ year $^{-1}$ regardless the frequency and duration of measurements.

v) We understand that especially in the arid climate, a time-scale of at least 3 years would be reasonable. However, data in this time-span were until now not provided by any other research project. Such frequent measurement as in our study are unique and provide detailed insights on redistribution processes as have never been monitored before.

**Table A4.** Summary of the volume of redistributed sediment for the period of 7 months, according to area and disturbance type. $Vol_{exc}$ describes volume of the sediment excavated by the animals. $Vol_{affected}$ describes volume of the sediment redistributed during rainfall events within affected areas. $Vol_{add}$ describes the difference in redistributed sediment volume within affected and not affected area during rainfall.

| Disturbance | Area | PdA | LC |
|---|---|---|---|
| $Vol_{exc}$ | Affected area | $9.57$ cm$^3$ cm$^{-2}$ 7 months$^{-1}$ | $8.53$ cm$^3$ cm$^{-2}$ 7 months$^{-1}$ |
| | Per burrow | $874.22$ cm$^3$ burrow$^{-1}$ 7 months$^{-1}$ | $715.52$ cm$^3$ burrow$^{-1}$ 7 months$^{-1}$ |
| | Hillslope-wide | $0.11$ m$^3$ ha$^{-1}$ 7 months$^{-1}$ | $0.39$ m$^3$ ha$^{-1}$ 7 months$^{-1}$ |
| $Vol_{affected}$ | Affected area | $-1.15$ cm$^3$ cm$^{-2}$ 7 months$^{-1}$ | $-6.09$ cm$^3$ cm$^{-2}$ 7 months$^{-1}$ |
| | Per burrow | $-73.71$ cm$^3$ burrow$^{-1}$ 7 months$^{-1}$ | $-511.22$ cm$^3$ burrow$^{-1}$ 7 months$^{-1}$ |
| | Hillslope-wide | $-0.03$ m$^3$ ha$^{-1}$ 7 months$^{-1}$ | $-0.28$ m$^3$ ha$^{-1}$ 7 months$^{-1}$ |
| $Vol_{add}$ | Affected area | $-0.69$ cm$^3$ cm$^{-2}$ 7 months$^{-1}$ | $-4.30$ cm$^3$ cm$^{-2}$ 7 months$^{-1}$ |
| | Per burrow | $-28.21$ cm$^3$ burrow$^{-1}$ 7 months$^{-1}$ | $-361.20$ cm$^3$ burrow$^{-1}$ 7 months$^{-1}$ |
| | Hillslope-wide | $-0.01$ m$^3$ ha$^{-1}$ 7 months$^{-1}$ | $-0.2$ m$^3$ ha$^{-1}$ 7 months$^{-1}$ |

**[R2C26]:**

Chapter 3.8: Did the authors perform any validation of their up-scaled data, e.g. by leaving out some samples for testing?

**[R2R26]:**

Yes, we applied Leave-One-Out cross validation and changed the manuscript accordingly:

Lines 387-388: "The model was validated by a Leave-One-Out cross validation."

**[R2C27]:**

Figure 4: Please, also state the standard deviation because it looks high according to the scatterplot.

**[R2R27]:**

We calculated the standard deviation and included the value in the figure (now Figure A3), Line 688.

**[R2C28]:**

Line 590-592: Why would more sporadic measurements be less reliable? The cumulative signal can be more significant than the more frequent measurements with smaller signal to noise ratio.

**[R2R28]:**

Thank you for raising your concern. The frequent measurements were necessary to answer our research question, namely to understand the dynamics of rainfall- and animal-caused sediment redistribution. As these two processes are alternating, less frequent measurements might miss sediment excavation process by the animal leading to the underestimation of the cumulatively redistributed sediment volume. For this, we rather accepted the possible error due to data noise as we found it to be much lower than the sediment volume which is excavated by the 
[revised manuscript text omitted]

---

## Author Response (AR2)

Dear editor,

Thank you for the positive feedback. We revised the manuscript according to your comments and provide a point-by-point response.

Kind regards

Paulina Grigusova

Dear Authors,

Thank you for submitting your revised manuscript and responses to the reviewers' comments. I found both reviewers to be quite positive regarding the scientific significance of your work. I also found your manuscript to present a methodologically interesting advancement in studying how animal burrowing cause sediment redistribution. Because of the extensive nature of the revisions, I would like to send out your manuscript for review again. But first, please address the following minor to moderate points:

• L86: Here and throughout the manuscript, the German 'und' is used instead of 'and' between two cited authors.

We corrected the reference style as requested.

• Throughout the manuscript: In addition to the above error, there are several small typographical errors with missing periods or odd word choices. I have commented on a few of these below and the manuscript will be subjected to language editing if accepted, but please take the time to do another careful read-through of the manuscript before submitting.

We reread the manuscript and corrected the typographical errors.

• Introduction: I agree with Reviewer #2 that a description of a low-cost ToF is needed. Although you have added a description in the second to last paragraph of the Discussion and expanded the description in the Methods section, the manuscript is still lacking a clear and concise description of what ToF photogrammetry is in the Introduction. Furthermore, what gap or problem from other types of photogrammetry does ToF fill. You describe this in the Methods, but it would be helpful to have the advantage of the system introduced in the Introduction to understand the significance of the work.

Thank you for this comment. We expanded the paragraph in the introduction as follows (Lines 122-127):

The Time-of-Flight (ToF) technology offers here a new possibility for surface monitoring, as a technique for a cost-effective high-resolution monitoring of sediment redistribution (Eitel et al., 2011; Hänsel et al., 2016) which can be achieved by a simple installation of one devise in the field is missing. ToF-based cameras illuminate the targeted object with a light source for a known amount of time and then estimate the distance between the camera and the object by measuring the time needed for the reflected light to reach the camera.

• L346-348: These sentences were added as a response to R2C22; however, I do not think that this completely answer the reviewer's question. The reviewer requested information on how you handled sediment redistribution caused by different processes, but the added text only describes how you may notice if there are several processes occurring. Please elaborate how you are able to parse out effects of different processes.

We explained the data processing in more detail (Lines 350-355):

If both animal-caused and rainfall-caused sediment redistribution took place, the following conditions applied: i) rainfall event occurred, ii) burrow size changed, iii) sediment eroded from not affected areas. Here, the animal-caused sediment redistribution was calculated as the sediment volume excavated from

the entrance. The rainfall-caused sediment redistribution was calculated as the sediment volume which eroded from the burrow roof and mound. To this amount we added the animal-caused redistributed sediment volume, as this sediment accumulated on the mound.

• L397-98: Given the authors' lengthy response to R2C25, I would like to see a condensed version of that response in the manuscript text to justify using the 7 months of data to extrapolate to one year. For example, I would highlight the high quality of the data used in this study, where there are in fact 7 months of data rather than only one event to extrapolate from, so this shows an advancement in precision of annual sediment redistribution.

We agree and we included the arguments in the manuscript (Lines 387-396):

Please note that we used the volume of redistributed sediment monitored for 7 months to calculate the volume of sediment per year. We decided to upscale due to several reasons: In contrast to previous studies, our study provides daily data on sediment redistribution which allow a more realistic temporal upscaling than the data sampling with lower frequency. All previous studies estimated the volume of redistributed sediment per year, even though the measurements were conducted less frequently (Table A6, A7 and A8) or even when the measurement was not repeated at all (Übernickel et al., 2021b). These studies thus completely ignored the ongoing sediment excavation and erosion processes. Our study was conducted from middle autumn to middle of spring and thus covered exactly half of the vertebrate burrowing season (Romanach et al. 2005), including dry and wet seasons, thus capturing the key cycles of variability.

• L590: I would recommend against a paragraph containing only two sentences. Combine this to another existing paragraph or expand on this idea.

We combined the paragraphs.

---

## Author Response (AR3)

Reviewer 1

Dear reviewer #1,

Thank you very much for your comments. We addressed all of your comments below and adjusted the manuscript accordingly.

A main concern from the previous round is that they speak of burrowed and not affected areas. The non-affected areas can be seen as controls. They should be spatially independent. I cannot imagine that this is the case within 3m2. Even with only one burrow in the 3m2, the area surrounding it, is probably not unaffected. Put another way, the sediment that moved outside of the burrow-affected area, should be sediment that also would have moved if there were no burrows, but at this spatial scale, I don't think you can draw a line beyond which the burrows don't affect sediment movement. The appropriate scale would have been a whole burrowed slope, compared to a slope with no burrows.

Thank you for raising this point. We now refer to the previously described "affected areas" as "burrows and to the previously "not affected areas" as "burrow embedding areas". We explained the definition of terms at Lines 288-290:

The remaining surface within the camera's FOV was burrow embedding area. Please note, that this area may still be affected by the burrowing activity of the animal and is not completely unaffected by the animal.

We rewrote the terms in the manuscript altogether 84 times all of which are marked green. We changed the description of figures 4, 5, 6, 7, A4, A5, A7, A8 and A9. We added your suggestion to compare hillslopes with and without burrows in the conclusion.

I also wanted to warn against circular logic in terms of distinguishing burrow affected sites. The response variable is changes in topography, but they also use topography to some extent to delineate burrows and non-burrows. This is similar to looking at the effects of invertebrates on soil nutrients and picking the invertebrate sites based on high nutrients. Again, having a whole slope of burrows and no burrows would negate this.

Thank you raising this point. Before any analysis, we delineated the areas roof, entrance, mound and adjusting areas. The delineation was based solely on the first saved frame of every camera. Each voxel was assigned to an area. The same voxel was part of the firstly assigned area in all of the remaining frames. For example, if parts of the mound eroded over time, the corresponding voxels were still parts of the area mound – thus, the amount of eroded sediment from mound could be calculated.

My biggest concern, however, is still the temporal upscaling to a year. I understand that other studies did not monitor continuously, but with biological organisms, which have phenological cycles, you need snapshots throughout the year. Animals go into hibernation in winter or torpor in summer, where they basically become completely inactive. There is nothing included here on the biology of these species (in fact, I am still not sure who the burrowers on these specific slopes are – is it all of the animals listed in the intro?). It is similar to measuring flowering rates in spring and then upscaling that to a whole year. Plants generally flower in spring, so you would greatly be diluting the rate by upscaling. Similarly, here, almost half of the year has not been included. I am not convinced that the animals carried on digging at the same rate.

We removed the temporal upscaling from the manuscript. Now, whenever we talk about absolute data, we use the amount of redistributed sediment for the time period of 7 months. The changes mainly affected abstract, lines 382-405, Table 1 and Figure 7. By discussing our results to the previous studies, we compare the relative changes and not the absolute changes.

Lines 49-52:

Abstract: The animal-caused cumulative sediment redistribution was 8.52 $cm^3 cm^{-2}$ 7 $months^{-1}$ in the mediterranean and 9.57 $cm^3 cm^{-2}$ 7 $months^{-1}$ in the arid climate zone. The rainfall-caused cumulative sediment redistribution within burrow was higher (-6.09 $cm^3 cm^{-2}$ 7 $months^{-1}$) in the mediterranean than the arid climate zone (-0.82 $cm^3 cm^{-2}$ 7 $months^{-1}$).

Lines 510-514:

Table 1. Summary of the volume of redistributed sediment, according to area and disturbance type. Volexc describes volume of the sediment excavated by the animals. Volburrow describes volume of the sediment redistributed during rainfall events within burrows. Voladd describes the difference in redistributed sediment volume within burrows and burrow embedding areas during rainfall.

| Disturbance | Area | PdA | LC |
|---|---|---|---|
| Volexc | Burrow | 9.57 cm3 cm-2 7 months-1 | 8.53 cm3 cm-2 7 months-1 |
| | Per burrow | 874.22 cm3 burrow-1 7 months-1 | 715.52 cm3 burrow-1 7 months-1 |
| | Hillslope-wide | 0.11 m3 ha-1 7 months-1 | 0.39 m3 ha-1 7 months-1 |
| Volburrow | Burrow | -1.15 cm3 cm-2 7 months-1 | -6.09 cm3 cm-2 7 months-1 |
| | Per burrow | -73.71 cm3 burrow-1 7 months-1 | -511.22 cm3 burrow-1 7 months-1 |
| | Hillslope-wide | -0.03 m3 ha-1 7 months-1 | -0.28 m3 ha-1 7 months-1 |
| Voladd | Burrow | -0.69 cm3 cm -2 7 months-1 | -4.30 cm3 cm-2 7 months-1 |
| | Per burrow | -28.21 cm3 burrow-1 7 months-1 | -361.20 cm3 burrow-1 7 months-1 |
| | Hillslope-wide | -0.01 m3 ha-1 7 months-1 | -0.2 m3 ha-1 7 months-1 |

Lines 574-576: Sediment redistribution within burrow areas was 40% higher at the arid research site, and at the mediterranean research site, it was 338% higher when compared to burrow embedding area

Lines 601-603: Our results indicate an up to 338% increase in the sediment volume redistributed during rainfall events measured within burrows when compared to burrow embedding areas. In contrast to our result, the maximum increase estimated in previous studies was 208%.

Lines 615-617: These studies estimated an increase in the volume of sediment redistributed during rainfall events, measured within burrows when compared to burrow embedding areas, to be between 205% and 473%.

Reviewer 2

The authors answered most of my comments very satisfactorily and changed the manuscript accordingly. However, three comments (two by myself and one by the editor) remain to which I think, some more explanation should be given:

Thank you very much for your comments and the positive feedback. We answered all of your remaining comments below.

Comment to [R2R2]: You might also add that ToF exhibits lower spatial resolution and areal coverage compared to time-lapse photogrammetry, but therefore can also be used at night as it is an active remote sensing tool and that the processing is less complex compared to photogrammetry because you immediately receive distance values in a local coordinate system.

We added the requested sentence to the manuscript:

Lines 123-126: In contrast, The Time-of-Flight (ToF) technology exhibits lower spatial resolution and aerial coverage compared to time-lapse photogrammetry. However, as an active remote sensing tool it can also be used at night. Additionally, the processing is less complex compared to photogrammetry because the distance values are immediately received in a local coordinate system.

I am afraid that I am then still not understanding the processing entirely because according to point [R2R10] you perform your analysis in a XY-plane with a height value for each pixel (which would be 2.5D). Why is a rigid body transformation not possible in your case? How is your approach different from the rigid body transformation?

Thank you for the inquiry. Rigid transformation describes rotations and reflections of objects which preserve the Euclidean space between every pair of points. We, however, had to correct the frames due

to errors caused by hillslope inclination and the inclination of the camera. Due to hillslope and camera inclination, the distance between points increased with increasing distance from the camera in the uncorrected dataset, Thus, we couldn't only rotate the object. If we would have only rotate the object, the parts of the burrow located farer from the camera would incorrectly be larger than in reality.

Comment to response to editor comment L346-348 (These sentences were added as a response to R2C22; however, I do not think that this completely answer the reviewer's question. The reviewer requested information on how you handled sediment redistribution caused by different processes, but the added text only describes how you may notice if there are several processes occurring. Please elaborate how you are able to parse out effects of different processes. I am still not able to understand, how the authors are able to ensure that sediment in the entrance moved due to animal activity and not rainfall:

We apologize for the misunderstanding. We extended the part explaining the calculations:

Lines 354-365: To attribute sediment redistribution to rainfall event, three preconditions had to be met: (i) A rainfall event occurred; (ii) sediment is eroded from burrow roof, mound and the embedding area; (iii) sediment is accumulated within the burrow entrance.

To attribute sediment redistribution to a combination of animal activity and rainfall, four preconditions had to be met: (i) A rainfall event occurred; (ii) sediment is eroded from embedding area; (iii) the height of burrow roof and mound decreased or increased; (iv) the depth of burrow entrance increased.

The animal-caused sediment redistribution was calculated as the sediment volume excavated from the entrance. Animal excavation always increased depth of the burrow entrance. The rainfall-caused sediment redistribution was calculated as the sediment volume which eroded from the burrow roof and mound. During a rainfall event, sediment eroding from burrow roof might accumulate within burrow entrances. In this case, the depth of the burrow entrance decreased. No sediment could erode from the entrance during a rainfall event. Decreased depth of a burrow entrance always points to sediment redistribution caused by rainfall, increased depth of burrow entrance always means redistribution by animals. Rainfall-caused redistribution always occurred before animal-caused redistribution, as without erosion caused by rainfall, the animals did not need to reconstruct their burrows.

Line 124: "which can be achieved by a simple installation of one devise in the field is missing" – is missing should be removed.

We removed "is missing" from the sentence.

Lines 126-128: ToF offers here a new possibility for surface monitoring, as a technique for a cost-effective high-resolution monitoring of sediment redistribution (Eitel et al., 2011; Hänsel et al., 2016) which can be achieved by a simple installation of one device in the field.

---

## Author Response (AR4)

Dear editor,

We implemented following changes in our manuscript.

We changed the rates back to units per year instead of 7 months.

We included a broad section about local burrowing animals (Lines 165 – 187):

**2.1 Local burrowing animals**

[revised manuscript text omitted]

---

## Author Response (AR5)

Dear editorial team, dear Lina Polvi Sjölberg,

We adjusted expended and adjusted the discussion part and calculated uncertainties regarding our usage of annual rate (Lines 597 – 624):

We have found that rainfall plays a key role in triggering burrowing activity, which means that wet seasons experience higher sediment redistribution rates than dry seasons. In the year of investigation (2019), the dry season lasted from January until April, and from September until December (8 months), and the wet season lasted from May until August (4 months). The monitoring period lasted from March until October which covered 3 dry and 4 wet months (7 months in total). A yearly rate of sediment redistribution can be calculated by simply averaging the redistribution rate of the 7 monitored months and multiplying this result by 12 months, which results in an average redistribution rate of 0.4 $m^2$ $ha^{-1}$ $year^{-1}$ for LC and 0.1 $m^2$ $ha^{-1}$ $year^{-1}$ for PdA. However, because burrowing activity and rain-driven sediment redistribution is mainly determined by rainfall, this method might have led to an overestimation of the annual redistribution rate based on averaging, because the unmonitored part of the year 2019 was predominantly dry (Übernickel et al. 2021a). This can be accounted for by adding five times the dry month redistribution rate to the monitored 7 months, which leads to a lower annual redistribution rates for LC of 0.3 $m^2$ $ha^{-1}$ $year^{-1}$ and for PdA of 0.1 $m^2$ $ha^{-1}$ $year^{-1}$. This difference between both values (0.1 $m^2$ $ha^{-1}$ $year^{-1}$ for LC and under 0.1 $m^2$ $ha^{-1}$ $year^{-1}$ for PdA) can be interpreted as the uncertainty range for the year of observation. However, decadal rainfall variability indicates that the year of monitoring (2019) was among the drier years of the last 30 years (Yáñez et al. 2001) which means our results might underestimate sediment redistribution on a longer time perspective.

Furthermore, the phenology of the burrowing animals is an additional source for uncertainty when calculating annual rates. The most common burrowing animal families in the area are active from March until October (refer to section earlier), and hence their burrowing activity is fully covered during the monitoring period. None of the most common burrowing animal families were reported to be active from November until February. This is also in line with our observations, because burrowing intensity increased from March until May, reached its peak between May and June and declined until September (Figure 6). By extrapolating from 7 months to one-year period, our estimated excavation was 0.7 $m^2$ $ha^{-1}$ $year^{-1}$ in LC and 0.8 $m^2$ $ha^{-1}$ $year^{-1}$ in PdA. By adding five times the low active months to the 7 months of observation, the estimated excavation would be 0.6 $m^2$ $ha^{-1}$ $year^{-1}$ in LC and 0.6 $m^2$ $ha^{-1}$ $year^{-1}$ in PdA. The excavation uncertainty range is thus 0.1 $m^2$ $ha^{-1}$ $year^{-1}$ for LC and 0.2 $m^2$ $ha^{-1}$ $year^{-1}$ for PdA. In summary, the discussion on the uncertainties of extrapolating single or sub-annual observations to annual rates clearly underpins the importance of high resolution, longer-term monitoring, which can be warranted with the here developed technology.

Kind regards
Paulina Grigusova

---

## Author Response (AR6)

Dear Lina,

We apologize for the misunderstanding during our last revisions. We added a discussion section regarding uncertainties caused by climate variability and phenology (Lines 597 – 625):

**5.2 The role of climate variability and burrowing cycles**

We have found that rainfall plays a key role in triggering burrowing activity, which means that wet seasons experience higher sediment redistribution rates than dry seasons. In the year of investigation (2019), the dry season lasted from January until April, and from September until December (8 months), and the wet season lasted from May until August (4 months). The monitoring period lasted from March until October which covered 3 dry and 4 wet months (7 months in total). A yearly rate of sediment redistribution can be calculated by simply averaging the redistribution rate of the 7 monitored months and multiplying this result by 12 months, which results in an average redistribution rate of 0.4 $m^2$ $ha^{-1}$ $year^{-1}$ for LC and 0.1 $m^2$ $ha^{-1}$ $year^{-1}$ for PdA. However, because burrowing activity and rain-driven sediment redistribution is mainly determined by rainfall, this method might have led to an overestimation of the annual redistribution rate based on averaging, because the unmonitored part of the year 2019 was predominantly dry (Übernickel et al. 2021a). This can be accounted for by adding five times the dry month redistribution rate to the monitored 7 months, which leads to a lower annual redistribution rates for LC of 0.3 $m^2$ $ha^{-1}$ $year^{-1}$ and for PdA of 0.1 $m^2$ $ha^{-1}$ $year^{-1}$. Our values might thus overestimate sediment redistribution for the year 2019. This difference between both values (0.1 $m^2$ $ha^{-1}$ $year^{-1}$ for LC and under 0.1 $m^2$ $ha^{-1}$ $year^{-1}$ for PdA) can be interpreted as the uncertainty range for the year of observation.

However, decadal rainfall variability indicates that the year of monitoring (2019) was among the drier years of the last 30 years (Yáñez et al. 2001; Valdés-Pineda et al. 2016; Garreaud et al. 2002; Wilcox et al. 2016). The amount of precipitation since 1980 ranges from 200 mm until 800 mm per year (https://climatologia.meteochile.gob.cl/application/requerimiento/producto/RE3005) while the amount of precipitation in 2019 was just above 100 mm. This means, our results might underestimate sediment redistribution on a longer time perspective by 2 - 7 times.

Furthermore, the phenology of the burrowing animals is an additional source for uncertainty when calculating annual rates. The most common burrowing animal families in the area are active for three months of the year. The months in which they are active, are between April and September. None of the most common burrowing animal families were reported to be active from November until February. (Eccard und Herde 2013; Jimenez et al. 1992; Katzman et al. 2018; Malizia 1998; Monteverde und Piudo 2011). This is also in line with our observations, because burrowing intensity increased from March until May, reached its peak between May and June and declined until September (Figure 6). By extrapolating from 7 months to one-year period, our estimated excavation was 0.7 $m^2$ $ha^{-1}$ $year^{-1}$ in LC and 0.8 $m^2$ $ha^{-1}$ $year^{-1}$ in PdA. By adding five times the low active months to the 7 months of observation, the estimated excavation would be 0.6 $m^2$ $ha^{-1}$ $year^{-1}$ in LC and 0.6 $m^2$ $ha^{-1}$ $year^{-1}$ in PdA. Our values might thus overestimate the sediment excavation and the excavation uncertainty range is 0.1 $m^2$ $ha^{-1}$ $year^{-1}$ for LC and 0.2 $m^2$ $ha^{-1}$ $year^{-1}$ for PdA.

Following the changes in discussion, we rephrased one of the paragraphs in the introduction (105 – 114).

The reason for this knowledge gap is that previous studies have not provided data on low magnitude but frequently occurring sediment redistribution due to a lack of spatio-temporal high-resolution microtopographic surface monitoring techniques which can also measure continuously in the field. Field experiments with, for example, rainfall simulators can unveil processes but cannot cover the time-dependant natural dynamics of sediment redistribution. When using erosion pins or splash boards, the sites had to be revisited each time and the data were thus obtained only sporadically (Imeson und Kwaad 1976; Hazelhoff et al. 1981; Richards und Humphreys 2010). This limited all previous studies in their explanatory power, because biotic-driven processes are typically characterised by small quantity and a frequent re-occurrence (Larsen et al. 2021). It is hence likely that previous studies based on non-continuously conducted measurements or rainfall experiments underestimated the role of burrowing animals on rates of hillslope sediment flux.

If possible, we would like to change the title from:

Time-Of-Flight based monitoring reveals higher sediment redistribution rates related to burrowing animals than previously assumed

to

Higher sediment redistribution rates related to burrowing animals as revealed by Time-Of-Flight based monitoring

as we feel, that due to the discussion regarding uncertainties, it might be misleading to clearly state in the title that our estimated sediment redistribution is "higher than previously assumed."

We also enhanced the description of some of our figures explaining positive and negative values, as we thought that the values might be misunderstood by the readers otherwise (Lines 488, 511, 516):

Positive values indicate sediment accumulation. Negative values indicate sediment erosion.

For the same reason, we added the words "excavation" and "erosion" in abstract (Lines 49 – 51):

The animal-caused cumulative sediment excavation was 14.6 $cm^3\ cm^{-2}\ year^{-1}$ in the Mediterranean, and 16.4 $cm^3\ cm^{-2}\ year^{-1}$ in the arid climate zone. The rainfall-caused cumulative sediment erosion within burrows was higher (10.4 $cm^3\ cm^{-2}\ year^{-1}$) in the Mediterranean than the arid climate zone (1.4 $cm^3\ cm^{-2}\ year^{-1}$).

Kind regards

Paulina Grigusova